# Partial asynchrony of coniferous forest carbon sources and sinks at the intra-annual time scale

As major terrestrial carbon sinks, forests play an important role in mitigating climate change. The relationship between the seasonal uptake of carbon and its allocation to woody biomass remains poorly understood, leaving a significant gap in our capacity to predict carbon sequestration by forests. Here, we compare the intra-annual dynamics of carbon fluxes and wood formation across the Northern hemisphere, from carbon assimilation and the formation of non-structural carbon compounds to their incorporation in woody tissues. We show temporally coupled seasonal peaks of carbon assimilation (GPP) and wood cell differentiation, while the two processes are substantially decoupled during off-peak periods. Peaks of cambial activity occur substantially earlier compared to GPP, suggesting the buffer role of non-structural carbohydrates between the processes of carbon assimilation and allocation to wood. Our findings suggest that high-resolution seasonal data of ecosystem carbon fluxes, wood formation and the associated physiological processes may reduce uncertainties in carbon source-sink relationships at different spatial scales, from stand to ecosystem levels.

In the Northern Hemisphere, the atmospheric $CO_2$ concentration undergoes a seasonal annual cycle ranging from 6 to 19 ppm, a net result of intra-annual fluxes greater than those from annual anthropogenic emissions[1]. This cycle is driven by the seasonal dynamics of terrestrial plant activity, i.e., the balance between the two main processes determining net accumulation of carbon (C) in the biosphere: gross primary production (GPP) and ecosystem respiration (RECO). GPP represents the total amount of $CO_2$ assimilated by the ecosystem, the result of the photosynthetic process. RECO represents the release of $CO_2$ to the atmosphere by oxidation of C compounds in vegetation and soil.

Once absorbed by leaves, atmospheric C undergoes a set of chemical reactions that lead to the formation of C compounds[2], a process known as C assimilation. The assimilated C is used as fundamental bricks to support plant growth and sustain the consequent respiratory costs. Among all plant tissues, wood is the main reservoir for long-term C sequestration, and its formation represents a highly C-demanding process[3]. During wood formation, C resources supply energy for cell division, contribute to generate and maintain turgor pressure during cell expansion, and are used to build complex compounds required for cell wall thickening and lignification[4]. During this process, a substantial fraction of C is released again into the atmosphere through respiration[5,6], a fraction thought to comprise up to 40% of total respiration from vegetation[7].

Historically, scientists have considered photosynthesis as a main driver of stem growth (source limitation)[3]. However, in recent decades, several studies have suggested that direct environmental and developmental constraints control wood growth (sink limitation)[8,9]. Nowadays, the extent to which stem growth is linked to carbon assimilation and their respective temporal relationships are still not completely understood. The temporal relationships between C assimilation and biomass production are the key to solve one of the largest sources of uncertainty in global vegetation models because the feedbacks of processes between terrestrial ecosystems and atmosphere are at the basis of the global C cycle[3]. Moreover, environmental drivers, mainly temperature and precipitations, affect C assimilation and wood

e-mail: roberto.silvestro1@uqac.ca

formation in different ways[10]. Thus, climate change might alter the C partitioning to the stem and affect the potential of net C sequestration in wood and the productivity of forests. For this reason, improvements in understanding the chronological sequence of growth phenological events and associated processes are crucial for a better understanding of C sequestration in woody tissues.

Previous meta-analyses have shown that northern conifers synchronize the activity of meristems with local climate, concentrating the timings of wood formation within precise time windows[11] when the environmental conditions are favourable for growth[11,12]. Similarly to wood formation, ecosystem C fluxes in forests have clear seasonal patterns according to thermal and precipitation gradients[13]. Several studies have tried to disentangle the temporal and functional relationship between ecosystem productivity and forest biomass production[14]. Nowadays, the possibility to use direct measurements and multi-year records of ecosystem C fluxes from eddy-covariance (EC) stations in forests has greatly increased the potential of assessing the association between C fluxes at ecosystem scale with wood production. However, these analyses have proposed different or even contrasting conclusions. While some studies found strong correlations between source and sink activities[15–20], others lacked in significant results[21–25].

The contrasting results reported by the literature in the last decades[15–25] could be explained by the different approaches employed to address the complex issue of assessing temporal relationships between these processes. In this context, some studies have explored temporal relationships between source and sink activities, focusing on interannual patterns. At a global scale, in particular, eddy covariance GPP records have been shown to be largely decoupled from tree growth at the inter-annual time scale[23]. Several reasons were proposed to explain this asynchrony, e.g., stored carbohydrates may provide much of the C necessary for growth during certain growth stages, and the seasonal dynamics of GPP and wood formation may substantially differ from one another. Nevertheless, it is crucial to highlight the importance of precisely defining the temporal resolution when investigating the temporal dynamics of specific processes. Indeed, studies conducted at annual resolution cannot assess physiological processes occurring during seasonal growth and the underlying mechanisms may significantly vary within and across years. This issue should be addressed with observations performed at higher temporal resolution because intra-annual growth-related physiological processes may demonstrate a buffering effect, possibly desynchronizing source and sink activities. A detailed analysis of these processes and the examination of their seasonal patterns at intra-annual scale may provide a more comprehensive understanding of the C cycle in forest ecosystems and quantify the degree of synchrony between C sources and sinks at different spatial scales, from stand to ecosystem.

This study aims to provide a comparative analysis of the intra-annual dynamics of C fluxes and wood formation, from C assimilation and the formation of non-structural compounds to their incorporation in woody tissues in conifers of the Northern hemisphere. Specifically, we generated a new database combining intra-annual data of (1) ecosystem-scale NEE (Net Ecosystem Exchange, the measure of net exchange of C between an ecosystem and the atmosphere per unit ground area[26]), GPP and RECO, (2) non-structural carbohydrates (NSC) concentrations in various tissues (needles, stems, roots), and (3) observations of cambial activity (i.e. cambial cell division) and wood formation (i.e. xylem cell differentiation) of 39 conifer species in boreal, temperate and Mediterranean biomes. We use this dataset to: (1) identify and describe the seasonal patterns of these processes; (2) assess the co-occurrence of seasonal peak and off-peak periods among processes; and, specifically, (3) determine the temporal relationships between C assimilation and wood formation at intra-annual scale. Given the high C-demanding nature of wood formation, we

hypothesize that a synchronization should exist between the seasonal peaks in carbon assimilation and cell differentiation during wood formation.

Our study reveals a synchrony between the seasonal peaks of carbon assimilation and wood formation in coniferous forests across the Northern Hemisphere. However, while carbon assimilation and cell differentiation are temporally coupled during peak periods, they are substantially decoupled during off-peak periods. These findings emphasize the importance of high-resolution seasonal data to reduce uncertainties in modelling forest carbon source-sink dynamics and enhance our understanding of carbon sequestration processes.

## Results and discussion
### Bioclimatic analysis
The dataset used in this work consists of observations collected at 177 sites belonging to boreal, temperate and Mediterranean biomes of the Northern hemisphere (Fig. 1).

Bioclimatic analyses were conducted to establish a more precise climate-based classification of the study sites. However, since this bioclimatic analysis did not significantly enhance the subsequent outcomes, we have chosen to present, in this text, the results based on the classification according to the biome to which each site belongs (i.e., boreal, temperate, and Mediterranean). Detailed results of the bioclimatic classification and the subsequent analyses following this classification are provided in Supplementary Note 1.

### Sources-sinks seasonal dynamics within and among biomes
The seasonal dynamics of wood formation (i.e., cambial activity, cell enlargement, and cell wall thickening and lignification), NSC (i.e., starch and soluble sugars in needles, stem and roots), and carbon fluxes (i.e., NEE, GPP and RECO) were fitted with skewed normal distribution or V-type exponential curves. All curves were significant (at least $p < 0.05$) with a residual standard error (RSE) ranging between 0.09 and 0.27 (Supplementary Note 3−Supplementary Table 3).

Seasonal peaks in C fluxes, NSC concentrations and number of cells during wood formation occur first in the Mediterranean biome, and later in the boreal biome (Fig. 2 right-hand panel). Soluble sugars in roots represent the only exception in the Mediterranean biome, where the minimum concentration is substantially later than that in boreal and temperate biomes.

The boreal biome is characterized by very sharp peaks, concentrated in a short time window of almost 60 days in which the climatic conditions are favourable for C assimilation and the development of tissues[11]. All processes peak in the boreal biome from mid-May to mid-July (Fig. 2). The growing season in the temperate biome last 73 days, from the beginning of May to the end of July (Fig. 2). The Mediterranean biome has the longest growing season, from the beginning of April to the beginning of October, for a total of 170 days (Fig. 2).

Within each biome, we observe a similar sequence of events during the growing season (Fig. 2 left-hand panel). At the beginning of the growing season, the peaks of NEE and cambial activity are followed after 1 week by peaks of starch content in needles and stems, and often roots. The peaks in starch content are followed by peaks in cell enlargement. Subsequently, we observe in quick succession the peaks of GPP, RECO and cell wall thickening and lignification. The peak in cell enlargement precedes GPP by 13 days. The peak of GPP is followed by peaks of cell wall thickening and lignification by 9 days, and peaks of RECO by 12 days. Finally, we observe a sequential event of minimum concentrations of soluble sugars in all organs (i.e., needles, stem and roots) (Fig. 2). Details of sampling and statistical fits are provided in Supplementary Methods and Supplementary Note 2.

NEE reached a maximum 32 days before GPP, and 44 days before RECO (Fig. 2). Our results generalize the evidence from local studies that NEE cumulates in spring, when the temperature and ecosystem

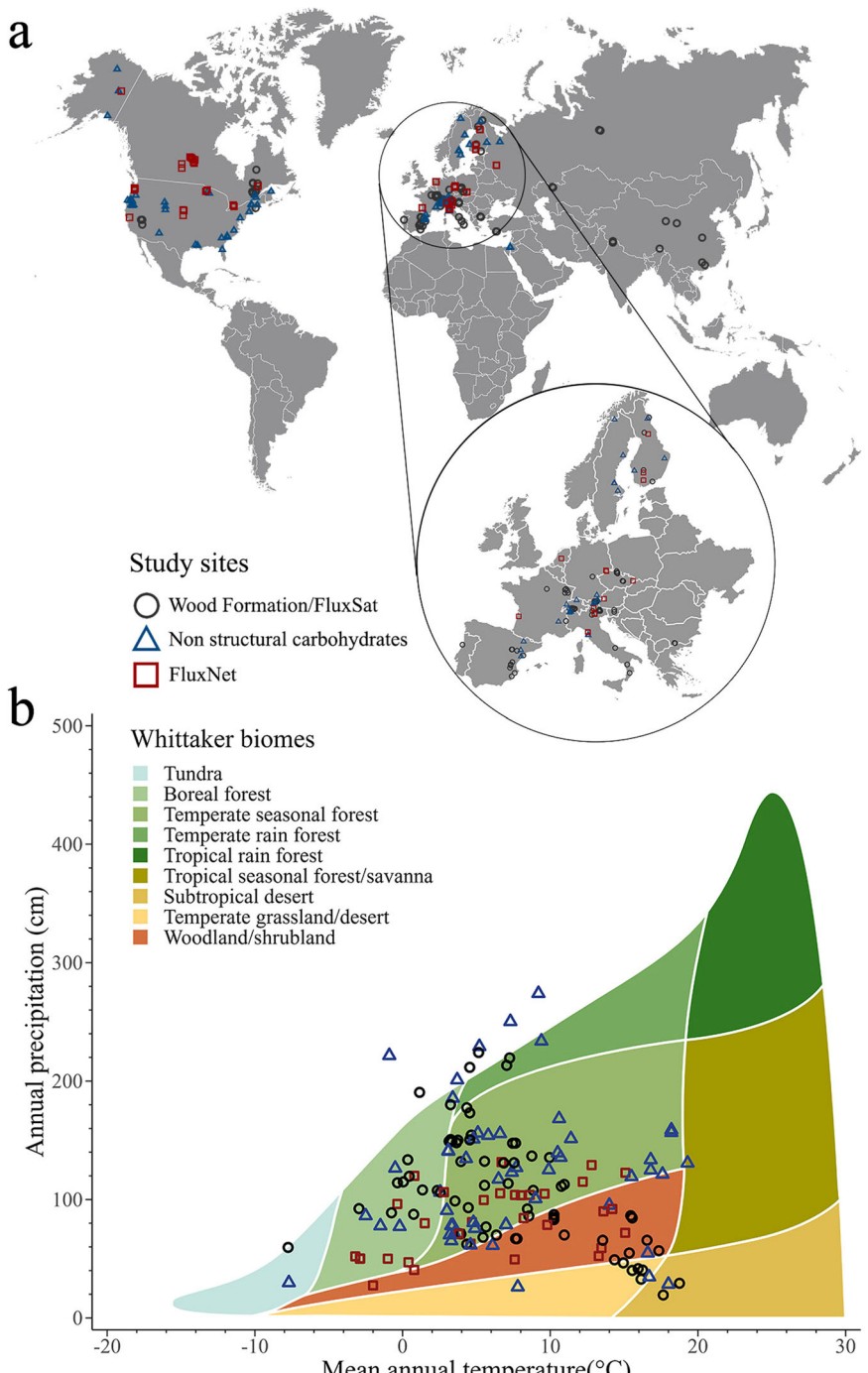

**Fig. 1 | Study sites and climatic characterization. a** Study sites in the Northern hemisphere where the dynamics of wood formation (81 sites), non-structural carbohydrates (57 sites) and C fluxes (39 sites) were determined. C fluxes were also estimated from the FluxSat product for each pixel corresponding to the 87 sites where wood formation was determined. **b** A Whittaker biome plot illustrates the mean annual temperature (°C) and mean annual precipitation (cm) for all 177 study sites. The figure was created using the R packages rnaturalearth, plotbiomes, and ggplot2.

respiration are still low[27]. When respiration culminates in the summer, the photosynthetic rate is reduced by high vapour pressure deficit[28], which explains the short time gap between GPP and RECO. For this reason, the culmination of GPP and RECO are associated to reductions in NEE[27]. Our results suggest that NEE is unable to define the timing of long-term C storage during wood formation.

 Assessment of the seasonal dynamics suggests that NSC may act as a buffer between C flux and the process of wood formation. The seasonal dynamics in NSC and their peaks in the different organs define two distinct periods of the growing season. A first period (i.e.,

early or late spring depending on the biome), characterized by a starch accumulation in plant organs lasting 51 days (average across biomes) (Fig. 2). A second period (i.e., late spring or early summer depending on the biome), with low concentrations of soluble sugars, lasting 21 days (average across biomes) (Fig. 2). Starch is a pure storage compound that may be severely depleted during tree growth, under harsh winter conditions, or after disturbance events (e.g., defoliation)[29,30]. During the tree lifespan, starch plays a dual role in C allocation[31]. Beyond its role as a long-term storage compound (especially in roots, tubers or seeds), starch can act as a sink in the form of a

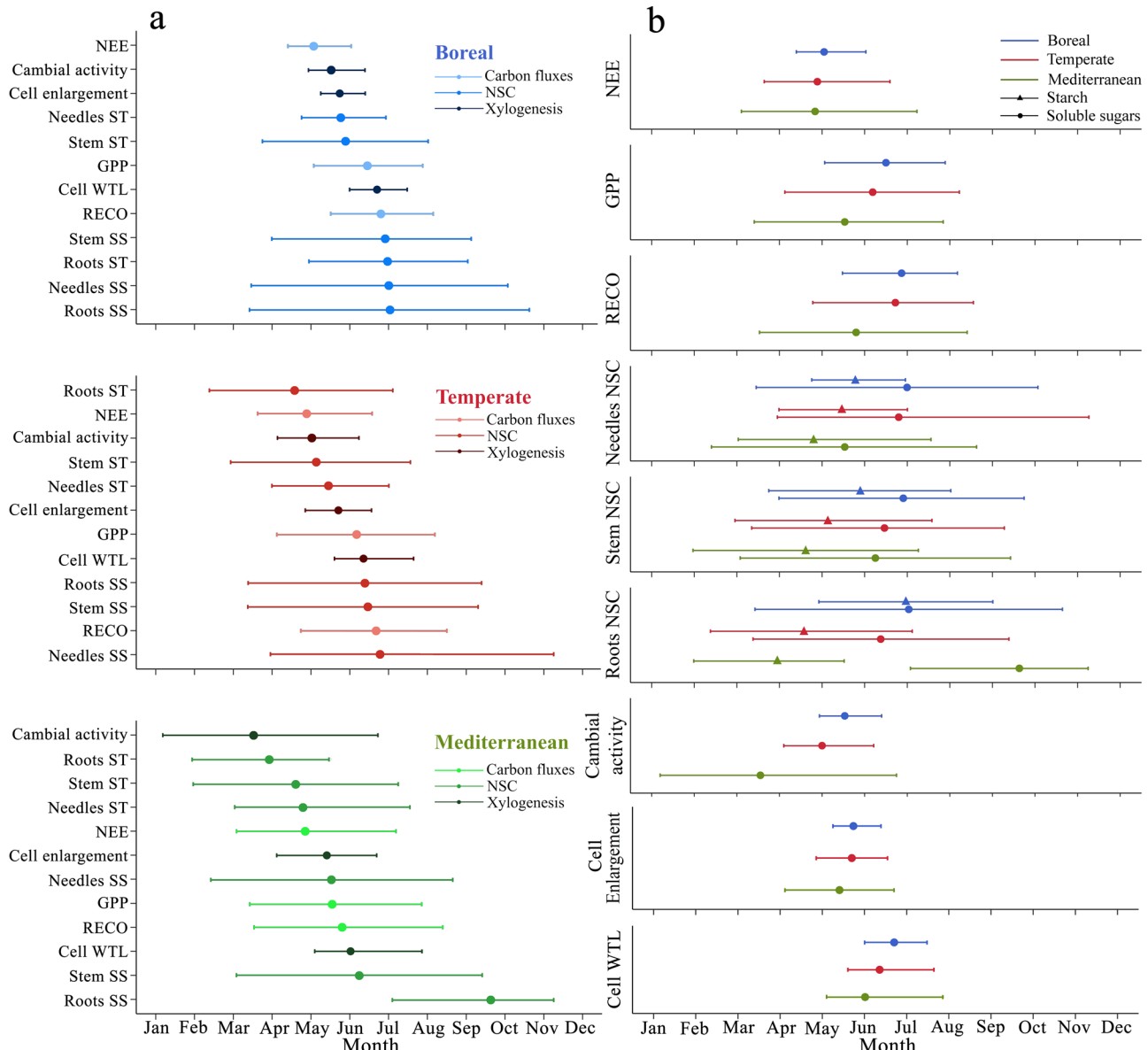

**Fig. 2 | Timing and duration of peak activity for carbon fluxes, NSC dynamics, and wood formation phases across biomes. a** Peak (dot or triangle) and period of maximum activity, the 75th percentile (horizontal error bars) for C fluxes, NSC dynamics and wood formation phenological phases. Minimum concentration is shown for soluble sugars. **b** for each biome, the sequence of temporal consecutive events during the growing season is shown. On the right, the difference among biomes for each event is shown. NEE, GPP and RECO represent the Net Ecosystem Exchange, Gross Primary Production and Ecosystem Respiration. ST and SS represent starch and soluble sugars, respectively. Cell WTL represents the phenological stage of cell wall thickening and lignification.

temporary C reserve[31]. Starch is converted to soluble sugars in winter to promote cold tolerance[32], and re-converted into starch at the beginning of the growing season[30]. Unlike starch, soluble sugars perform various immediate functions, including the supply for new growth, metabolism, osmoregulation, defence, and adopting the role of intermediary metabolites and substrates for transport[33]. Soluble sugars, therefore, need to remain consistently above a physiological threshold[2,34].

**Cambial activity and C assimilation**
The peak of cambial activity in trees is uncoupled from the peaks of photosynthesis and ecosystem respiration (Figs. 3 and 4). Standardized major axis (SMA) regressions were all significant ($p < 0.05$, $R^2$ ranging from 0.21 to 0.81) (Table 1). Cambial activity peaked 30–60 days earlier than GPP, in the boreal and Mediterranean biome, respectively (Fig. 4). Moreover, the slopes of these relationships

differed significantly among biomes, ranging from 0.51 in the boreal biome to 0.98 in the Mediterranean biome (Table 1).

The highest rates of xylem cell division occur at the onset of the growing season, which explains the asynchrony between the peaks of cambial activity and GPP. The beginning of cell division corresponds with the peak of starch content (Fig. 2) and the reactivation of primary growth, i.e. bud burst[35]. The relationship between primary and secondary growth is regulated by a physiological trade-off in which the two processes depend on one another but are also certainly in competition for the same resources[4,35]. In this phase, developing buds represent a priority sink, supported mainly by new sugars produced with the photosynthesis occurring in old needles. In contrast, cambial cell division also relies on reserves stored in the stem rather than C translocated from the leaves[4,29,36,37]. During the growing season, the peak in cambial cell division represents a watershed moment in which the prioritization of resources changes to be finally committed to

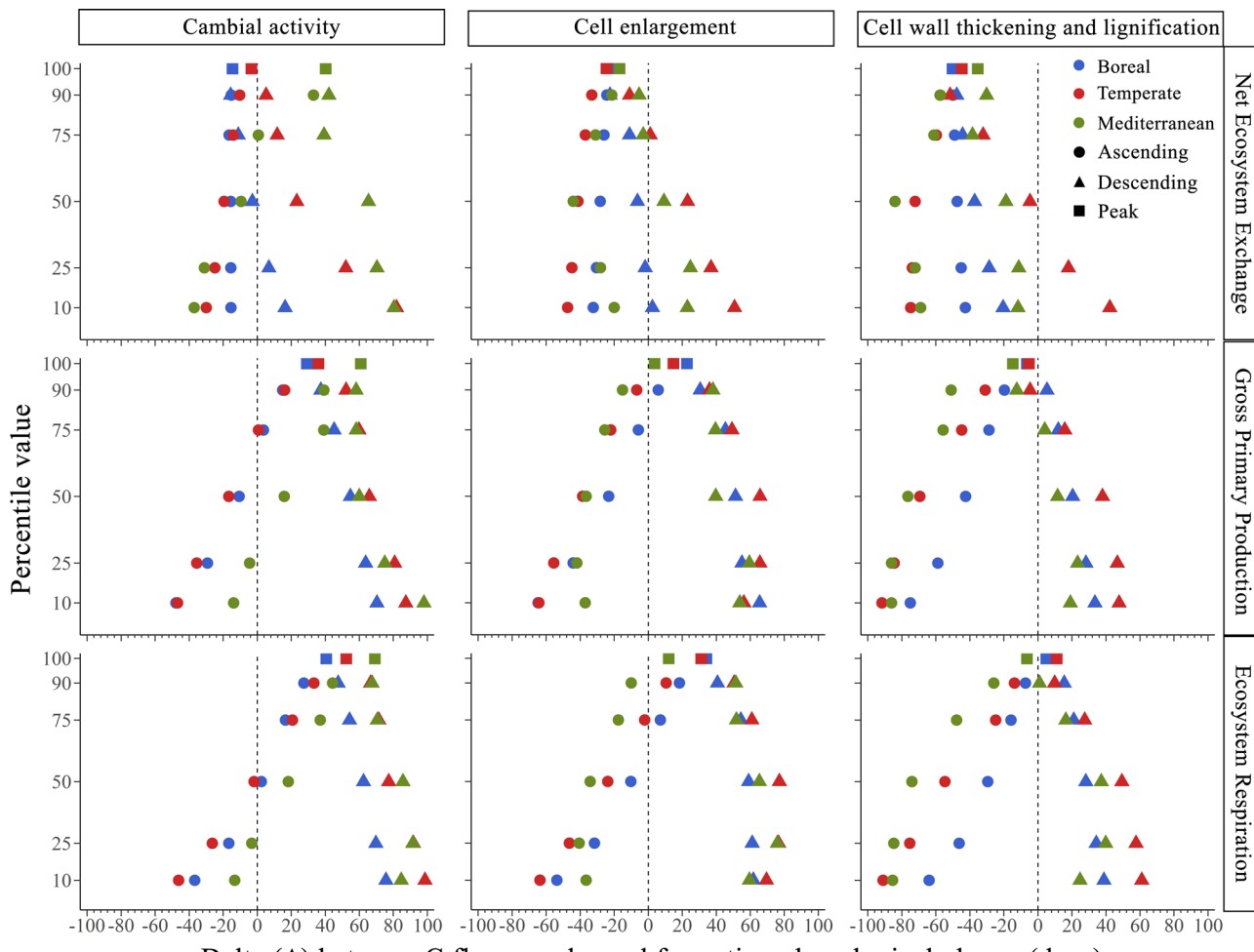

**Fig. 3 | Differences in timing between carbon fluxes and wood formation phases across biomes.** Differences (i.e., subtraction) between the timing of different percentiles and peak (i.e., 100th percentile) of C fluxes, i.e., NEE, GPP, RECO and those of phenological phases of wood formation, i.e., cambial activity, cell enlargement and cell wall thickening and lignification in boreal, temperate and Mediterranean biomes. Negative deltas indicate that C fluxes occurred earlier, while positive deltas that C fluxes occurred later, relative to wood formation, for each percentile shown in the figure.

secondary growth. It is likely that the peak of cambial activity is reached when sufficient resources have been allocated to support cell differentiation, which may account for the coordination between peaks in cell division and accumulation in reserves (i.e. starch). Several regulation and post-translational processes are known to control the allocation to starch as well as the enzymatic hydrolysis of soluble sugars[31].

**The sequence of C allocation in wood**
The peak of xylem cell differentiation is temporally coupled with the peak in photosynthesis and ecosystem respiration, suggesting a close relationship between C assimilation and allocation, and related respiratory costs (Fig. 3). In all biomes, the culmination of GPP occurs 13 days later than the peak of cell enlargement, and 9 days earlier than the peak of wall thickening and lignification. The same results were obtained when the timing of cell differentiation phenological phases was compared with the estimated GPP extracted from FluxSat (Fig. 4). The regressions comparing the timing of the peak of GPP and both cell enlargement and cell wall thickening and lignification phenological phase were all significant ($R^2 = 0.21$–0.43 for cell enlargement, and $R^2 = 0.31$–0.81 for cell wall thickening and lignification, both $p < 0.05$) (Table 1). In both phenological phases, the regressions presented a common slope across biomes of 0.78 for cell enlargement and 0.80 for cell wall thickening and lignification. Given that the slopes were not

significantly different, we tested for differences in intercept (Table 1). The intercepts were statistically different across biomes, with the boreal biome showing the earliest peaks in both cell enlargement and cell wall thickening and lignification (Table 1 and Fig. 4). The Mediterranean biome showed the latest peaks.

Photosynthesis and cell differentiation are synchronized across a wide spatial scale (Fig. 3), probably because the two processes occur during the time window when environmental conditions are optimal for both. By considering the 10th percentiles of the fitted curves, the onset of photosynthesis occurs earlier (55 days, average across biomes) than the onset of cell differentiation during wood formation (i.e., onset of cell enlargement stage) (Fig. 3, Supplementary Table 7). Conversely, considering the 10th percentiles of the descending portions of the curve the ending of photosynthesis occurs later (33 days, average across biomes) than the ending of cell differentiation (i.e., ending of cell wall thickening and lignification stage) (Fig. 3, Supplementary Table 7). As we progress towards the peaks of these processes, the time lag between GPP and wood formation gradually diminishes (Fig. 3, Supplementary Table 7). During the period of maximum activity (75th percentile, ascending portion of the curves), the time gap between GPP and the cell enlargement stage narrows to an average of 18 days across biomes, and 12 days across biomes between GPP and the cell wall thickening and lignification stage, in the descending portions of the curve (Fig. 3, Supplementary Table 7). This

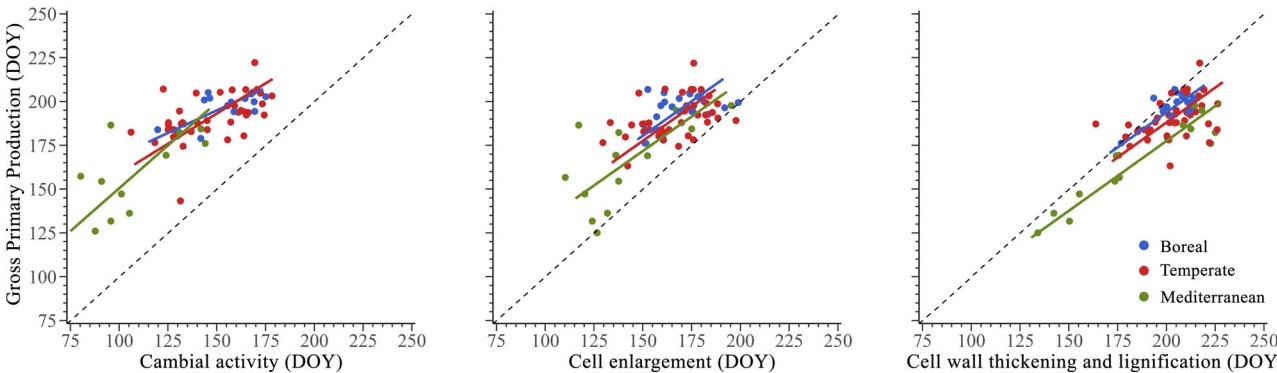

**Fig. 4 | Synchrony of peaks between GPP and wood formation phases across biomes.** Synchronisms among the timing of peak of Gross Primary Production (GPP) and phenological phases of wood formation (i.e., cambial activity, cell enlargement, and cell wall thickening and lignification) in 81 sites across boreal, temperate and Mediterranean biomes. The dashed line represents a bisecting line (1:1).

**Table 1 | Results of Standardized Major Axis (SMA) analyses of the bivariate relationships among timing of the peak of GPP and timing of cambial activity, cell enlargement and cell wall thickening and lignification in 81 sites across boreal, temperate and Mediterranean biomes**

| Phenological stage | Biome | Y-intercept | Slope | 95% CI slope | $R^2$ |
|---|---|---|---|---|---|
| Cambial activity | Boreal | 118.01 | 0.51 | 0.34 - 0.77 | 0.50 |
| | Temperate | 89.83 | 0.69 | 0.52 - 0.91 | 0.27 |
| | Mediterranean | 52.71 | 0.98 | 0.56 - 1.71 | 0.40 |
| Cell enlargement | Boreal | 62.64 | 0.78 | 0.63 – 0.96 | 0.26 |
| | Temperate | 59.74 | 0.78 | 0.63 – 0.96 | 0.43 |
| | Mediterranean | 53.93 | 0.78 | 0.63 – 0.96 | 0.21 |
| Cell wall thickening and lignification | Boreal | 11.16 | 0.8 | 0.69 – 0.94 | 0.51 |
| | Temperate | 23.73 | 0.8 | 0.69 – 0.94 | 0.31 |
| | Mediterranean | 28.26 | 0.8 | 0.69 – 0.94 | 0.81 |

asynchrony for the onset and ending of these processes is linked to the different sensitivities of these processes to environmental drivers[9,38]. Indeed, it is well known that growth-related processes, such as cell enlargement and the synthesis of cell wall and proteins are more sensitive to temperature and water stress than photosynthesis[9,39].

Photosynthesis is less constrained by environmental factors than meristematic activity and cell differentiation. In conifers, photosynthesis can occur as soon as leaf temperature is above freezing point, and the tree has sufficient water availability[40]. Moreover, photosynthesis can stop and reactivate according to changes in the weather[41,42]. On the contrary, the flexibility observed in C assimilation cannot be found in primary and secondary meristematic activity. These processes are triggered by a specific set of environmental cues that act as signals for reactivation during precise time windows, ensuring optimal growth while minimizing the risks of damage[11,12,43,44]. Unlike photosynthesis and the resulting C assimilation, the growth process cannot be completely stopped or resumed based on sudden changes in environmental factors during the growing season.

The peak of GPP falls between the preceding peak in cell enlargement (averaging 13 days across biomes) and the subsequent peak in cell wall thickening and lignification (averaging 8 days across biomes) (Fig. 3). Cell division and cell enlargement are known to be turgor-driven processes, while secondary-wall formation is based on the supply of sugars[45]. In summer, once primary growth (i.e. shoot elongation) is completed, secondary growth (i.e. xylem formation) can benefit from a strong and continuous supply of carbohydrates[45]. At the peak of xylem development, specifically during latewood formation, the phloem pool acts as the ultimate C source for wood formation[46,47]. Accordingly, the growth process prevalently uses C assimilated during the current growing season[46–49]. This synchronism between sink and source could explain the culmination of GPP between the peaks of cell enlargement and secondary wall formation: the moment the demand for C is higher, the supply is also greater.

## Variance and predictors of phenological events

To determine whether the distribution of available data for the main processes of ecosystem fluxes and wood formation affected our conclusions, we used random forest regression models to assess the relative importance of study year, site, species and biome as predictors for the peak timing of NEE, GPP and RECO. These variables explained between 24 and 44% of the variance. Overall, the $R^2$ ranged between 0.72 and 0.96 for the training set and 0.49 to 0.68 for the test set (Supplementary Table 9). Biome resulted as the most important predictor followed by site and study year across all models (Supplementary Fig. 14). We observed the same pattern in the random forest model applied for FluxSat data (Fig. 5), where the model explained 54.06% of the variance, showing an $R^2$ of 0.89 for the training set and 0.66 for the test set (Supplementary Table 9).

In the random forest regression models for the peak timing of cambial activity, cell enlargement, and cell wall thickening and lignification phases, the species of the monitored trees was also considered as a predictor alongside study year, site, and biome. These models explained from the 38.09 to the 43.09% of the variance, with $R^2$ ranging from 0.76 to 0.83 for the training set and from 0.70 to 0.83 for the test set (Supplementary Note 3; Supplementary Table 8). The importance of the biome was confirmed, followed by site, species, and study year in each model (Fig. 5). However, in the model for cambial activity, the species showed a greater importance than the site (Fig. 5).

Biome emerged as the most influential predictor for the peak timing of both wood formation processes and ecosystem C fluxes. This

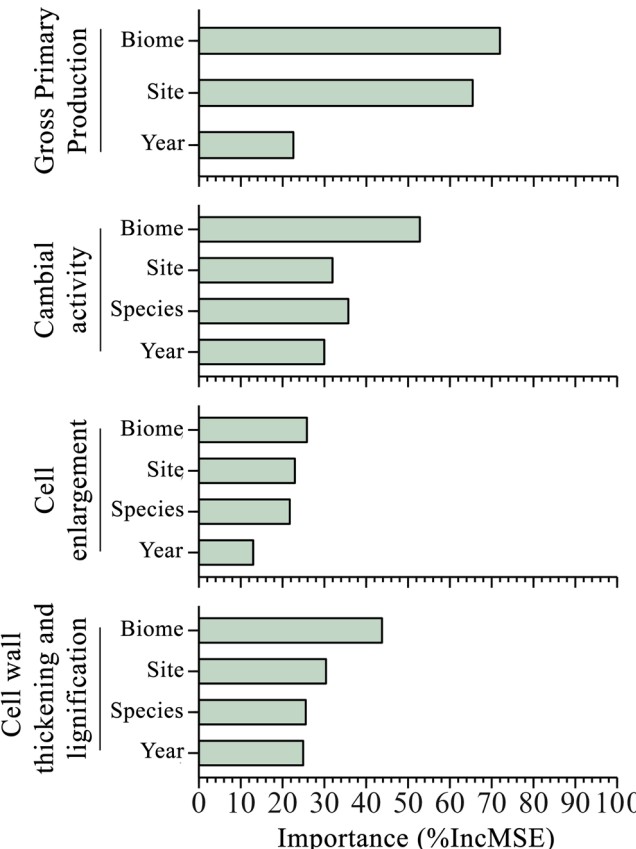

**Fig. 5 | Importance of predictors in random forest models for GPP and wood formation phenology.** Relative importance in terms of Mean Decrease Accuracy (%IncMSE) of predictors in the random forest regression models for the timing of peak of FluxSat Gross Primary Production (GPP) and phenological phases of wood formation (i.e., cambial activity, cell enlargement, and cell wall thickening and lignification).

result underscores the substantial role of the broader climatic context in shaping the temporal dynamics in source and sink activities. The observation that site exceeds the importance of the study year suggests a more important influence of site-specific environmental conditions in determining the temporal occurrence of seasonal peaks. This possibly implies a more conservative pattern of peak occurrences, calibrated to the local characteristics, rather than a response to annual whether variations.

A prior study focusing on conifers in cold environments showed that the rate of xylem cell production culminates around the summer solstice[50]. After that date, cell production gradually decreases until ceasing. This pattern suggests that trees would have evolved by synchronizing their growth rates with day length[50]. Conversely, growth reactivation (i.e., reactivation of secondary meristem) and onset of xylem cell differentiation, despite the variability within populations[51,52], are driven by weather conditions, mainly temperature and water availability[4,6,11,12,43,44].

These insights not only clarify the outcomes of our random forest regression model but substantiate the observed synchronism of peaks in ecosystem C fluxes and cell differentiation, contrasting their asynchrony during the onset and ending of the growing season. Indeed, photosynthesis experiences fewer constraints from environmental factors compared to meristematic activity and cell differentiation[11,41,42], resulting in the desynchronization of both onset and ending of source and sinks activities. However, considering that both the rates of photosynthesis[53] and xylem cell production[50] respond to day length, it is plausible that this factor predominantly governs the synchronization

of source and sink peaks. Day length likely acts as a constant environmental factor over time, ensuring the convergence of a high demand with a proportionately high supply.

Finally, we emphasize the comparable significance of predictors between the site and the species. However, these conclusions are drawn from the analysis of phenological timings in conifers. Therefore, we cannot directly recognize the potential variation introduced by broadleaf species.

### Reconciling C-source and sink dynamics

The limited availability of sites with concurrent measurements of C fluxes, NSC concentrations, and wood phenology has hindered our ability to quantitatively explore the relationships between these three categories of processes. The extent to which C sources and sinks are closely linked is fundamentally determined by the magnitude of C fluxes between different compartments, which are exceedingly difficult to measure. However, given the ongoing debates surrounding the significance of potential drivers, temporal sequences, and the level of correlation among these processes at larger temporal and spatial scales, we believe that our temporal analysis provides a unique broad picture of the overall coordination of source and sink activities (and ultimately of C sequestration in wood) at large scales. Consequently, our study serves as a foundation to deepen our understanding of the consequences of climate changes on C sequestration in Northern hemisphere forests, along with exploring potential biome-specific responses.

The asynchrony between C assimilation and cell differentiation during the wood formation process observed in many local studies, can be largely attributed to limiting carbon sinks governed by meristem activity (i.e., C sink limitation to growth)[8]. In isolation, the asynchrony between these two processes can be attributed to their respective different sensitivities to environmental factors and possibly to differences in biome-specific resource use strategies. At broader scales, however, our results suggest that this asynchrony may originate from the longer time window of C assimilation compared to the period of C allocation to woody tissues. This discrepancy is also evident in the periods of maximum activity of the processes, as demonstrated by the present study.

The different responses of photosynthesis and growth to the climate may also account for the asynchronies observed at local scale, e.g., under drought conditions. Water deficit can decouple growth from photosynthesis[6,23], and likely explain the greater variability in the peaks of photosynthesis and cell differentiation observed in the Mediterranean biome in this study (Fig. 4). As a typical response to water deficit, C allocation (i.e., growth) always decreases before C assimilation (i.e., photosynthesis)[6,9,39].

In Mediterranean regions, a potential lag between C assimilation and xylem formation can be bimodal, as tree radial growth may slow down in response to water shortage, and resume in autumn after summer suspension[6,54]. These conditions cause a rise in the concentration of C stores, resulting in the accumulation of compatible solutes (typically hexoses and polysaccharides) in sink organs[6]. These compounds serve as energy sources and as sources of compatible sugars and other carbon-based solutes, protect subcellular compartments against the harmful effects of water loss, and increase drought tolerance[6]. This process of C accumulation in sink organs helps to explain how water deficit uncouples growth from photosynthesis. However, in precipitation manipulation experiments an almost synchronic coupling between new assimilates and xylem formation was observed when trees were irrigated after a long drought period[55]. This suggests that the observation of an asynchrony between photosynthetic activity and growth is primarily a response to limiting environmental conditions for growth.

Environmental constraints limit sink activity, which may be the primary cause of decoupling between the processes of C assimilation

and growth. However, C-sink limitation should not be interpreted as a complete disjunction between C resources and sink activity. C is a limited resource and, in the form of soluble sugars and starch, fulfils various functional roles, such as transport, energy metabolism, osmoregulation, and provides substrates for the synthesis of defence, exchange and biomass components[33]. While the prioritization of certain uses of newly-formed photoassimilates is not well understood, there is evidence that processes related to survival have a higher priority than increasing structural biomass, i.e., growth[47]. This is supported by evidence showing that trees prioritize storage even when experiencing a C shortage[56]. Therefore, it is also possible that, beyond direct environmental limitations, the partial asynchrony found here between C fluxes and wood formation may reflect varying allocation priorities among different species and biomes, reflecting different functional traits and resource use strategies. In summary, direct environmental cues and biotic disturbances can limit sink activity and desynchronize C assimilation and growth. In addition, the competition among sinks for a limited resource should likely be expected to further contribute to this asynchrony.

Numerous sites are currently equipped for monitoring and measuring carbon fluxes in various forest ecosystems. However, within the last decade, only a small percentage of these sites have been monitoring wood formation on an intra-annual scale. Incorporating detailed intra-annual observations of NSC pools, which act as a buffer between atmospheric C flux measurements (e.g., eddy covariance GPP and RECO) and intra-annual assessments of forest biomass growth (i.e., wood formation monitoring and intra-ring analysis of wood anatomy), can facilitate the complex task of reconciling, and ultimately clarifying, the temporal and functional relationships between C uptake and long-term C sequestration. This, in turn, will enable the examination of the timing and magnitude of C transfer processes and C use efficiency across various spatial and temporal scales. Such an analysis is a crucial step to reduce the sources of uncertainty in global vegetation models and achieve a deeper understanding of the C cycle at global scale.

## Methods
### Data selection
This study used data collected in 177 sites of boreal, temperate and Mediterranean biomes across the Northern hemisphere. We used mean annual precipitation and mean annual temperature data for each site, to calculate the Whittaker biome classification of all sites using R package "plotbiomes"[57] (Fig. 1). The sites are located in North America, Europe and Asia, and distributed over latitudes from 23 to 68 °N and elevations reaching 3,850 m a.s.l. (Fig. 1). The sites were assigned to a specific biome based on information in the papers from which the data were extracted. The study covers 38 coniferous species belonging to eight genera (Supplementary Methods, Supplementary Table 2). The dataset consists of the temporal dynamics of wood formation, i.e. cambial activity and xylem differentiation (81 sites), non-structural carbohydrates, i.e. starch and soluble sugars in needles, stem and roots (57 sites), and C fluxes, i.e. Net Ecosystem Exchange (NEE), Ecosystem Respiration (RECO), and Gross Primary Production (GPP) (39 sites), this latter dataset extracted from the FLUXNET2015 dataset[58]. The observations collected in the sites where wood formation was monitored were combined with estimates of gross primary production (GPP) using modelled data from FluxSat v2.0[59]. FluxSat outputs consist of a gridded GPP product obtained by upscaling FLUXNET data using MODIS reflectance[59]. Details on site selection and assembly of the datasets are reported in Supplementary Methods, whiles site information, coordinates, and data sources are reported in Supplementary Data 1.

### Bioclimatic analyses
To assess the climatic differences among sites where wood formation has been monitored, we collected bioclimatic data from the CHELSA bioclimatic database V2.1[60] with a spatial resolution of 30 arcseconds. Out of the 19 available bioclimatic parameters[60], we selected seven variables (Supplementary Table 1), excluding those that provided overly general descriptions of climate (e.g., annual temperature and precipitation) and removing variables that were highly correlated with each other ($r > |0.7|$). Subsequently, to group the study sites based on their climate-related characteristics, we applied the Partitioning Around Medoids (PAM) clustering algorithm, which is an extension of the k-means clustering algorithm[61]. To determine the optimal number of clusters, we utilized the Within-Sum-of-Squares method (WSS), which minimizes the distance between points within each cluster. Therefore, we determined that four clusters were the optimal choice for grouping the wood formation study sites. We then conducted a Principal Component Analysis (PCA) on the bioclimatic variables to determine the climatic classification of our 81 sites (Supplementary Note 1). Pearson's correlation coefficient was employed to identify the climatic factors that influenced the ordering of wood formation study sites by principal components (Supplementary Table 1). Bioclimatic analyses have been performed in R version 4.2.2 and its "factomineR" and "factoextra" packages.

### Assessment of seasonal patterns
The seasonal patterns of wood formation (i.e., cambial activity and xylem cell differentiation), NSC concentrations (i.e., starch and soluble sugars in needles, stem and roots) and C flux (i.e., NEE, GPP, RECO) for each biome were determined by performing non-linear regressions on data normalised between 0 and 1. The normalisation was performed to reduce the variability within the same range and compare the temporal dynamics among sites. We applied two non-linear parametric functions that are a generalisation of the normal distribution, skewed normal distribution[62] and the V-type exponential curve[63]. The former allows for non-zero skewness, the latter is a symmetric curve that describes concave, convex and linear shapes. The skewed curve, enabling curve asymmetry, was preferred over the exponential curve, which was used only when data had a concave or linear-like shape not allowed by the first curve.

The skewed normal distribution curve[62] is given by the following formula:

$$\text{Seasonal pattern} = \frac{1}{\omega * \sqrt{2\pi}} * e^{\left(\frac{1}{2} * \left(\frac{t-\xi}{\omega}\right)^2\right) * \left(1 + \text{erf} * \left(\alpha * \left(\frac{t-\xi}{\omega * \sqrt{2}}\right)\right)\right)} \quad (1)$$

$$\text{erf}_z = \text{error function} = \frac{2}{\sqrt{\pi}} \int_0^z e^{-t^2} dt \quad (2)$$

Where $\xi$ represents the location parameter, which determines the "location" or shift of the distribution; $\omega$ represents the scale parameter; $\alpha$ represents the shape or form parameter. This parameter skews the normal distribution to the left or right. The time is represented by t, included as a month for NSC seasonal pattern and DOY (day of the year) for wood formation and C fluxes.

The V-type exponential curve[63] is given by the following formula:

$$\text{Seasonal pattern} = Y\max * \mu^{(t-X\max)^2} \quad (3)$$

Where $\mu$ controls the growth rate of the curve; parameter $Ymax$ controls the height (Y-value) of function maximum or minimum; parameter $Xmax$ controls the location (X-value) of the function maximum or minimum. The curve has a minimum for $Ymax > 0$ and $Xmax > 1$, and a maximum for $Ymax > 0$ and $Xmax < 1$. It is horizontal line (i.e., $Y = Ymax$) for $\mu = 1$. The time is represented by $t$, included as the month for NSC seasonal pattern and DOY (day of the year) for wood formation observations and C fluxes. Non-linear regressions have been performed in R version 4.2.2 using "nls.multistat" package.

Each curve estimated the timing of the maximum value (or minimum while considering soluble sugar concentrations). In addition to this, fitted curves pertaining to the seasonal pattern of wood formation (i.e., cambial activity, cell enlargement, and cell wall thickening and lignification) and carbon fluxes derived from FluxNet data (i.e., NEE, GPP, and RECO) were utilized to estimate the timing of key percentiles, namely the 10th, 25th, 50th, 75th and 90th percentiles, for both the ascending and descending portions of the curves (Supplementary Data 2). We opted to use the 75th percentile as the threshold for defining the period of maximum activity. However, when the curve exhibited a minimum in soluble sugar concentrations, the period of maximum activity was determined based on the 25th percentile. We assessed the area under each curve (AUC) and assessed the AUC of the maximum activity itself by means of definite integrals by considering the period of maximum activity interval (Supplementary Note 2).

To assess functional scaling among biomes, we assessed the bivariate relationships between the timings of culmination of the GPP extracted from FluxSat, and wood formation using standardized major axis (SMA). Global scaling patterns (i.e., intercepts and slopes ± 95% confidence intervals) were obtained from the fitted regressions. Slopes were compared between biomes using a likelihood ratio test[64]. When the biomes had similar slopes, we tested for intercept differences using a Wald test[64]. SMA regressions have been performed in R version 4.2.2 using "smatr" package.

To analyze the variability and influence of specific predictors on the timings of peaks of C fluxes and wood formation, we employed skewed normal distribution curves. These regressions delineated the seasonal patterns of cambial activity and xylem cell differentiation for each study year, species, site, and biome. The same methodology was applied to FluxNET (i.e., NEE, GPP, RECO) and FluxSat (i.e., GPP) data. For each regression, the timings of the maximum value (i.e., peak timing) of each process was extracted. A random forest regression model was utilized to quantify the relative importance of predictors in determining the peak timing for each process.

For each process, we split the timings of the maximum values into a training set (80%) and a test set (20%) to assess the model performance. Five-fold cross-validation with five repetitions was employed as a resampling method to ensure more robust performance metrics. The goodness of fit for the regression models was evaluated using the coefficient of determination ($R^2$) for both the training and test sets, while the root mean squared error (RMSE) was employed to measure the accuracy of the models. Random forest models have been performed in R version 4.2.2 using "randomForest" package.

### Reporting summary
Further information on research design is available in the Nature Portfolio Reporting Summary linked to this article.

## Data availability
Data generated in this study have been deposited in Borealis: https://doi.org/10.5683/SP3/JRDOU1. Wood formation raw data are available under restricted access and can be accessed directly contacting the corresponding author of this study or using the procedure in Borealis. Data on non-structural carbohydrates are available at Dryad: https://doi.org/10.5061/dryad.j6r5k. Access to FluxNet data is provided through the FluxNet portal: https://fluxnet.org/data/fluxnet2015-dataset/. FluxSat data can be accessed via the FluxSat portal: https://daac.ornl.gov/cgi-bin/dsviewer.pl?ds_id=1835. Details of all sites used for the non-structural carbohydrates data, FluxNet data, and geographical coordinates for downloading FluxSat data are listed in the Supplementary Methods.

## Code availability
Code generated in this study have been deposited in Borealis: https://doi.org/10.5683/SP3/JRDOU1.

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

## Acknowledgements

This work was funded by the Ministère des Forêts, de la Faune et des Parcs du Québec, Fonds de Recherche du Québec - Nature et Technologies (AccFor, project #309064), the Observatoire régional de recherche en forêt boréale and Forêt d'Enseignement et de Recherche Simoncouche. R.

Silvestro received the Merit scholarship for international PhD students (PBEEE) by the Fonds de Recherche du Québec - Nature et Technologies (FRQNT) and a scholarship for an internship by the Centre d'étude de la forêt (CEF) realized at the Centre for Ecological Research and Forestry Applications (CREAF). V. Shishov is appreciated for supporting the RSF project #22-14-00048. C.B.K. Rathgeber would like to thank the Agence Nationale de la Recherche (ANR) in the framework of the Investissements d'Avenir (ANR-11-LABX-0002-01, Laboratoire d'Excellence ARBRE) for the support given to its work as well as the SILVATECH platform (Silvatech, INRAE, 2018. Structural and functional analysis of tree and wood Facility, doi: 10.15454/1.5572400113627854E12) for its contribution to the acquisition of wood formation monitoring data. K. Čufar, J. Gričar and P. Prislan were funded by the Slovenian Research and Innovation Agency, ARIS, research core funding Nos.: P4-0430 and P4-0015, projects: J4-2541, J4-4541 and Z4-7318 while K. Čufar, V. Gryc, P. Prislan and H. Vavrčík were also funded by the European Union's Horizon 2020 research and innovation program ASFORCLIC under the grant agreement 36 N°952314. The study was supported by the Austrian Science Funds (FWF), project numbers P322203-B and DOC-171-B "The future of Mountain Forests". F. Biondi was funded, in part, by the Experiment Station of the College of Agriculture, Biotechnology, and Natural Resources at the University of Nevada, Reno (USA). P. Fonti and R. Peters were funded by the Swiss National Science Foundation through projects INTEGRAL (grant no. 121859, PF), LOTFOR (grant no. 150205) and CALEIDSCOPE (grant no. 212902, PF). A. Lintunen was funded by the European Union–NextGenerationEU instrument and the Research Council of Finland (grants 347782, 355142). H. Mäkinen received funding from the European Union—NextGenerationEU instrument and is funded by the Research Council of Finland (grant no. 347782). G. Drolet and J.-D. Sylvain were funded by the Ministère des Forêts, de la Faune et des Parcs of Quebec, Canada (Evap-for, project #142332139). The authors thank A. Garside for editing the English text. A. Giovannelli received funding from National Recovery and Resilience Plan (NRRP), Mission 4 Component 2 Investment 1.4—Call for tender No. 3138 of 16 December 2021, rectified by Decree n.3175 of 18 December 2021 of Italian Ministry of University and Research funded by the European Union–NextGenerationEU; Project code CN_00000033, Concession Decree No. 1034 of 17 June 2022 adopted by the Italian Ministry of University and Research, CUP B83C22002930006, Project title "National Biodiversity Future Center—NBFC".

## Author contributions

R.Silvestro, M.M., V.B. and S.R. conceived the ideas and designed methodology; S.A., A.A., F.B., J.J.C., F.C., H.Cochard., K.Č., H.E.Cuny., M.D.L., A.D., G.D., M.V.F., P.F., A.Giovannelli, J.G., A.Gruber, V.G., R.G., A.Güney, X.G., J.-G.H., T.J., J.K., A.V.K., T.K., A.Lemay, X.L., E.L., A.Lintunen, F.Liu, F.Lombardi, Q.M., H.Mäkinen, R.A.M., E.M.C., J.M.-V., S.M., H.Morin, C.N., P.N., W.O., J.M.O., A.P.O., T.V.S.P., M.P., R.L.P., P.R., P.P., C.B.K.R., A.Sala, A.Saracino, L.S., P.S.-A., V.V.S., A.Stokes, R.Sukumar, J.-D.S., R.T., V.T., J.U., H.V., J.V., G.A., Y.W., B.Y., Q.Z., S.Z., E.Z., and S.R. provided local wood formation and non-structural carbohydrates data; R.G.-V. downloaded and assembled FluxSat data; R.Silvestro and S.R. assembled the final datasets; R.Silvestro analysed data and led the writing of the manuscript. All authors contributed to the drafts and gave final approval for publication.

## Competing interests

The authors declare no competing interests.

## Additional information

Roberto Silvestro [1] ✉, Maurizio Mencuccini [2,3], Raúl García-Valdés[4,5], Serena Antonucci [6], Alberto Arzac[7], Franco Biondi [8], Valentina Buttò[1,9], J. Julio Camarero [10], Filipe Campelo [11], Hervé Cochard[12], Katarina Čufar [13], Henri E. Cuny[14], Martin de Luis [15], Annie Deslauriers[1], Guillaume Drolet[16], Marina V. Fonti[17], Patrick Fonti [17], Alessio Giovannelli[18], Jožica Gričar[19], Andreas Gruber[20], Vladimír Gryc[21], Rossella Guerrieri [2,22], Aylin Güney [23], Xiali Guo[24], Jian-Guo Huang [25], Tuula Jyske[26,27], Jakub Kašpar[28,29], Alexander V. Kirdyanov[7,30,31], Tamir Klein [32], Audrey Lemay[1], Xiaoxia Li[33], Eryuan Liang [33], Anna Lintunen[34,35], Feng Liu [36], Fabio Lombardi[37], Qianqian Ma[36,38], Harri Mäkinen [26], Rayees A. Malik [39,40], Edurne Martinez del Castillo [41], Jordi Martinez-Vilalta [2,42], Stefan Mayr [20], Hubert Morin[1], Cristina Nabais[11], Pekka Nöjd[26], Walter Oberhuber [20], José M. Olano[43], Andrew P. Ouimette[44], Teemu V. S. Paljakka [35], Mikko Peltoniemi [45], Richard L. Peters[17,46], Ping Ren[33,47], Peter Prislan [19], Cyrille B. K. Rathgeber [48], Anna Sala[49], Antonio Saracino [50], Luigi Saulino [50], Piia Schiestl-Aalto [34], Vladimir V. Shishov [7], Alexia Stokes[51], Raman Sukumar[39], Jean-Daniel Sylvain [16], Roberto Tognetti [52], Václav Treml[28],

**Josef Urban** ⓘ [7,53], **Hanuš Vavrčík** ⓘ [21], **Joana Vieira**[54], **Georg von Arx** ⓘ [17,55], **Yan Wang**[34,51], **Bao Yang** ⓘ [56], **Qiao Zeng**[57], **Shaokang Zhang**[36,38], **Emanuele Ziaco** ⓘ [41] **& Sergio Rossi**[1]

[1]Laboratoire sur les écosystemes terrestres boreaux, Département des Sciences Fondamentales, Université du Québec à Chicoutimi, 555 boulevard de l'Université, Chicoutimi, QC G7H2B1, Canada. [2]CREAF, E08193 Bellaterra (Cerdanyola del Vallès), Barcelona, Catalonia, Spain. [3]Institució Catalana de Recerca i Estudis Avançats (ICREA), Passeig de Lluis Companys 23, 08010 Barcelona, Spain. [4]Department of Biology and Geology, Physics and Inorganic Chemistry, Rey Juan Carlos University, c/ Tulipán s/n, 28933 Móstoles, Spain. [5]Global Change Research Institute (IICG-URJC), c/ Tulipán s/n, 28933 Móstoles, Spain. [6]Dipartimento di Agricoltura, Ambiente e Alimenti, Università degli Studi del Molise, 86100 Campobasso, Italy. [7]Siberian Federal University, 79 Svobodny pr., 660041 Krasnoyarsk, Russia. [8]DendroLab, Department of Natural Resources and Environmental Science, University of Nevada, Reno, NV 89557, USA. [9]Forest Research Institute, Université du Québec en Abitibi-Témiscamingue, Rouyn-Noranda, QC, Canada. [10]Instituto Pirenaico de Ecología, Consejo Superior de Investigaciones Científicas, 50192 Zaragoza, Spain. [11]Centre for Functional Ecology, Associate Laboratory TERRA, Department of Life Sciences, University of Coimbra, 3000-456 Coimbra, Portugal. [12]Université Clermont Auvergne, INRAE, PIAF, 63000 Clermont-Ferrand, France. [13]University of Ljubljana, Bio-technical Faculty, 1000 Ljubljana, Slovenia. [14]Institut National de l'Information Géographique et Forestière (IGN), 54250 Champigneulles, France. [15]Department of Geography and Regional Planning, Environmental Science Institute, University of Zaragoza, 50009 Zaragoza, Spain. [16]Direction de la Recherche Forestière, Ministère des Ressources Naturelles et des Forêts du Québec, 2700 rue Einstein, Québec, QC G1P 3W8, Canada. [17]Swiss Federal Institute for Forest, Snow and Landscape Research WSL, Zürcherstrasse 111, CH-8903 Birmensdorf, Switzerland. [18]Istituto di Ricerca sugli Ecosistemi Terrestri, Consiglio Nazionale delle Ricerche, 50019 Sesto Fiorentino, Italy. [19]Slovenian Forestry Institute, 1000 Ljubljana, Slovenia. [20]Department of Botany, Leopold-Franzens University of Innsbruck, 6020 Innsbruck, Austria. [21]Department of Wood Science and Wood Technology, Mendel University in Brno, 61300 Brno, Czech Republic. [22]Department of Agricultural and Food Sciences, University of Bologna, 40127 Bologna, Italy. [23]Izmir Katip Çelebi University, Faculty of Forestry, Izmir, Türkiye. [24]College of Forestry, Guangxi Key Laboratory of Forest Ecology and Conservation, Guangxi University, Daxue East Road 100, Nanning, Guangxi 530004, China. [25]MOE Key Laboratory of Biosystems Homeostasis and Protection, College of Life Sciences, Zhejiang University, Hangzhou 310058, China. [26]Natural Resources Institute Finland, Latokartanonkaari 9, 00790 Helsinki, Finland. [27]Department of Forest Sciences, University of Helsinki, PO Box 27 (Latokartanonkaari 7) 00014, Helsinki, Finland. [28]Department of Physical Geography and Geoecology, Charles University, CZ-12843 Prague, Czech Republic. [29]Department of Forest Ecology, The Silva Tarouca Research Institute for Landscape and Ornamental Gardening, Brno, Czechia. [30]V.N. Sukachev Institute of Forest SB RAS, Federal Research Center 'Krasnoyarsk Science Center SB RAS, 660036 Krasnoyarsk, Akademgorodok, Russia. [31]Department of Geography, University of Cambridge, Cambridge CB2 3EN, UK. [32]Department of Plant and Environmental Sciences, Weizmann Institute of Science, Rehovot 76100, Israel. [33]State Key Laboratory of Tibetan Plateau Earth System, Environment and Resources (TPESER), Institute of Tibetan Plateau Research, Chinese Academy of Sciences, Beijing 100101, China. [34]Institute for Atmospheric and Earth System Research / Physics, Faculty of Science, P.O. Box 68, University of Helsinki, FI-00014 Helsinki, Finland. [35]Institute for Atmospheric and Earth System Research / Forest Sciences, Faculty of Agriculture and Forestry, P.O. Box 27, University of Helsinki, FI-00014 Helsinki, Finland. [36]South China National Botanical Garden, Guangzhou 510650, China. [37]Dipartimento di Agraria, Università Mediterranea di Reggio Calabria, 89122 Reggio, Calabria, Italy. [38]Key Laboratory of Vegetation Restoration and Management of Degraded Ecosystems, Guangdong Provincial Key Laboratory of Applied Botany, South China Botanical Garden, Chinese Academy of Sciences, Guangzhou 510650, China. [39]Centre for Ecological Sciences, Indian Institute of Science (IISc), Bangalore 560012, India. [40]Department of Botany, University of Kashmir, India-190006 Kashmir, Srinagar, India. [41]Institute of Geography, Johannes Gutenberg University Mainz, Mainz, Germany. [42]Universitat Autònoma de Barcelona, Bellaterra (Cerdanyola del Vallès), E08193 Barcelona, Catalonia, Spain. [43]EIFAB, iuFOR. Universidad de Valladolid, Campus Duques de Soria, E-42004 Soria, Spain. [44]Earth Systems Research Center, Institute for the Study of Earth, Oceans, and Space, University of New Hampshire, Durham, NH, USA. [45]National Resources Institute (Luke), Helsinki, Finland. [46]Department of Environmental Sciences – Botany, University of Basel, Schönbeinstrasse 6, CH-4056 Basel, Switzerland. [47]College of Life Sciences, Anhui Normal University, Wuhu 241000, China. [48]Université de Lorraine, AgroParisTech, INRAE, SILVA, F-54000 Nancy, France. [49]Division of Biological Sciences, University of Montana, Missoula, MT, USA. [50]Department of Agricultural Sciences, University of Naples Federico II, I-80055 Portici, Napoli, Italy. [51]AMAP, University of Montpellier, CIRAD, CNRS, INRAE, IRD, Montpellier, France. [52]Faculty of Agricultural, Environmental and Food Sciences, Free University of Bozen-Bolzano, Piazza Università 5, 39100 Bozen-Bolzano, Italy. [53]Department of Forest Botany, Dendrology and Geobiocenology, Faculty of Forestry and Wood Technology, Mendel University in Brno, Zemedelska 1, 61300 Brno, Czech Republic. [54]CoLAB ForestWISE - Collaborative Laboratory for Integrated Forest & Fire Management, Quinta de Prados, 5000-801 Vila Real, Portugal. [55]Oeschger Centre for Climate Change Research, University of Bern, Hochschulstrasse 4, CH-3012 Bern, Switzerland. [56]School of Geography and Ocean Science, Nanjing University, Nanjing 210093, China. [57]Guangdong Open Laboratory of Geospatial Information Technology and Application, Guangzhou Institute of Geography, Guangdong Academy of Sciences, Guangzhou 510070, China. ✉e-mail: roberto.silvestro1@uqac.ca

