## [Peer Review File · Nature Communications]

Partial asynchrony of coniferous forest carbon sources and sinks at the intra-annual time scale.REVIEWER COMMENTS

Reviewer #1 (Remarks to the Author):

The authors combine flux tower CO₂ fluxes (GPP, RECO, NEE) in 39 sites, cambial activity in 85 sites and non-structural (NSC) carbohydrates in 57 sites representative of conifers in boreal, temperate, and Mediterranean forest stands to study the seasonal timing of plant processes, such as carbon fluxes, NSC dynamics, and wood formation. The study is motivated by the need to reconcile carbon sources and sinks controls on wood formation and therefore carbon sequestration with obvious implications for plant growth representation in terrestrial ecosystem models and global carbon cycle.

The manuscript is well written and presented, I read with a lot of interest. However, while I agree with most of what is discussed in the manuscript, I think the new results add very little and do not largely fulfill the risen expectations for a number of reasons detailed below.

1. The analysis of the diverse datasets that have been compiled is rather simplistic, the authors just search for the seasonal maximum of a given process/variable by fitting some non-linear function (LL 406-409) to normalize the time series. This reduces the new results to the timing of the peak of the various processes (Fig. 2, 3, 4), but with no information about their magnitude or the intra-seasonal variability of the process itself. While of course the knowledge of the DoY of maximum activity in so many case studies is relevant, it is only a very partial information. For instance, I am wondering, if it would not have been better to fit a CDF at least for some of the variables, and discuss the timing of the 10, 25, 50, 75, 90 percentiles, rather than the "maximum". It would have been more representative of the overall seasonality of the process; the maximum is really a specific information. The outcome of the simplistic analysis is that conclusions are remarking largely well known seasonal sequences as that NEE peaks before GPP or the well-known order of cambial activity: cell division, cell enlargement and cell wall thickening and lignification (reviewed by some of the authors themselves, Rathgeber et al 2016). As a matter of fact, most of these processes have a strong temperature signal and it is not surprising to see correlation across sites that have different average temperatures, the temperature effect itself will shift all of them later in the season in colder sites (Fig. 4). Most of what is discussed inside the article is correct, but it is a review based on literature and previous knowledge (e.g., LL 250-260, LL 323-330 and many others), with very little or no-addition from the results presented here. In other words, most of the article could have been written without the results of Fig. 2, 3 and 4.

2. The presented results are all based only on temporal sequences, but there is no assessment of the process magnitude. In my view, reporting values of the different fluxes in gC m⁻² day or storages in gC m⁻² is fundamental, if we want to advance the field and really address the challenges discussed by the authors (LL 162-166, 190-197, 380-381). I understand that for some variables (e.g., NSC dynamics, cambial activity) scaling values to forest stand might be particularly complex, but even with uncertainties we should aim at that. There is a value in knowing when certain processes are taking place in the growing season, however, it is their magnitude which controls the relative importance of sources and sink controls. One can have NPP and growth fully synchronized seasonally, but as wood growth is only a fraction of NPP (e.g., Wolf et al 2011), it is much more difficult to discuss their direct link. In other words, we need to understand seasonally where the gC m⁻² are allocated, stored, and translocated. For instance, I found the article of Gough et al 2009 focusing on a single site and estimating actual stand scale magnitudes of C allocation much more useful than looking at hundreds of sites in some normalized fashion. This is what I was expecting from the title and introduction of such an article.

3. Finally, the intra-annual perspective, if analyzed in terms of actual magnitude, can provide a lot of information on the mechanisms of carbon allocation, I agree with LL 188-190. However, when one wants to quantify carbon sequestration on wood biomass (a major motivation, LL 161-169), it is the correlation at the annual or multi-annual scale (if we account for previous year legacy and storage) between GPP and wood formation that should matter. If some C meant for wood is partially stored during the growing season and allocated to wood a few weeks later is surely relevant for process

understanding, but it will not modify the overall C budget and it will unlikely require substantial changes to current models.

Specific Comments.

P. 10 LL 332-333. I think this sentence is incorrect and a much more nuanced approach to the fact that CO₂ fertilization might induce considerable changes, in various processes, which have been widely documented (e.g., Walker et al 2021), should be made.

P. 11 LL 365. Here, the text contradicts with LL 332-333.

P. 11. LL 371-374 This is largely repetitive of what has been written before.

References

Rathgeber, C. B., Cuny, H. E., & Fonti, P. (2016). Biological basis of tree-ring formation: a crash course. *Frontiers in Plant Science*, 7, 734.

Gough C.M., Flower, C.E., Vogel, C.S., Dragoni, D., Curtis, P.S., 2009. Whole-ecosystem labile carbon production in a north temperate deciduous forest. *Agric. Forest Meteorol.* 149, 1531–1540, <http://dx.doi.org/10.1016/j.agrformet.2009.04.006>.

Walker, A. P., De Kauwe, M. G., Bastos, A., Belmecheri, S., Georgiou, K., Keeling, R. F., ... & Zuidema, P. A. (2021). Integrating the evidence for a terrestrial carbon sink caused by increasing atmospheric CO₂. *New phytologist*, 229(5), 2413-2445.

Wolf, A., Field, C. B., & Berry, J. A. (2011). Allometric growth and allocation in forests: a perspective from FLUXNET. *Ecological Applications*, 21(5), 1546-1556.

Reviewer #2 (Remarks to the Author):

This study is very novel and will certainly have a high impact in the field, since it fills a gap of understanding carbon (C) cycle with a multi-disciplinary approach considering all different factors around the seasonal C assimilation by plants.

The work was performed with conifers, but can in the future be expanded to angiosperms and help us understand forest C assimilation dynamics also in the tropics, which contain the largest biodiversity in the planet.

In this specific study, however, why were only conifers included? They represent a low diversity in terms of total trees in the planet, although of considerable biomass (Siberian Taiga.)

Also, for the future I think it would be invaluable to include conifers growing in tropical regions (in mountains, for instance). I think that would have given another twist to the work, since the climatic seasonality would have been completely different.

Some sentences, like the very first one from Intro, would benefit from citations.

I don't have any major criticism for the article, and I think it is a good fit for the journal.

Reviewer #3 (Remarks to the Author):

This study uses a dataset of seasonal components of vegetation productivity and wood formation to quantify variation in timing of these processes and obtain insights into the dependencies of these processes. Specifically, the study provides the insight that wood formation depends on sugars produced by photosynthesis in an indirect way. That is: the analyses of timing of these processes, plus those in starch, suggest that sugars produced are first stored before being used in wood formation. This analysis provides a valuable addition to the literature on source-sink limitation on woody growth, mostly because it illustrates how source and sink limitation vary over the season. The study is conducted at a remarkably large scale, in three NH climate zones.

I have four main comments:

1. The manuscript lacks clear questions or hypotheses on the expected timing of the studied processes. Instead, only objectives/aims are mentioned. At times, I got the impression that the study is quite descriptive. I would advise the authors to formulate hypotheses, e.g., on the order of processes, and differences between biomes. This will also help selecting results to be shown.
2. Sites and representativeness. A total of 180 sites is used, and for some variables (e.g., GPP) gridded products are used. Figure 1 shows there are hardly (no?) sites at which multiple processes (starch, wood formation, etc) are measured. This may have important implications for within-biome comparisons of different processes and cross-biome comparisons of the same process. For instance, if wood xylogenesis in biome X is measured at warmer sites, while starch levels at cooler sites, the difference in timing of these processes will not only be determined by the order of processes for an average tree in these sites, but also by across-site differences in climate. As far as I understand, this issue has not been accounted for. It is important to conduct a robustness test to verify whether climatic conditions within biomes differ across sites that deliver different processes. Similarly, differences in the years covered by the field studies included may also have an impact, which needs to be quantified. And, as far as I can tell, the analyses do not take into account site (or species) information as a random factor, and now mix data within and across sites. Finally, I think it would be important to show how representative sites are for the biomes studied. Is the full climate range represented?
3. In the main text, very little information is provided about the uncertainty of the analyses, or variation within and across biomes. No mention is made of the explained variance (or RSE) of the statistical models. Variation across sites is only shown in Figure 4, but not in 1-3. And in the text, information on peak days, or duration of processes is given without any information on variation, i.e. no standard deviations or errors or CI95 are provided.
4. I think the presentation of the results in Figures can be substantially improved. In Fig 1: the map could show forest cover in the background (or duration of growing season?), could be limited to NH sites, and it would have been good to plot sites in climate space as well, so show representativeness. IN Figure 2, the same information is provided on the left and right panels. Also a grouping of processes (related to photosynthesis, to storage or to wood formation) by colouring lines would be helpful. And, finally on this Figure, I thought that actual fitted curves in the SI were more informative than the way results are presented in Fig 2, because one can see the intensity of the processes over time. In Fig 3, I do not get what is shown, and the caption was not really helpful in explaining.

Two minor comments:

- In the text, the term mismatch is used, which -- to me -- suggests that something is going 'wrong', but that's clearly not the case. Processes are out of sync, but they are still connected functionally.
- The focus in the text and Figures is on comparing the peaks in the processes, but why not the onset of processes, or the time at which 90% of the seasonal total is reached? Is the analyses of peaks really the best way to get insights on seasonal dynamics of source-vs-sink limitation?

In the present file we provide detailed responses to reviewers' comments and feedback. We thank the reviewers for their meticulous revision efforts, which have significantly contributed to improving the clarity and coherence of the manuscript.

Below, you will find detailed responses to each comment and feedback received:

Reviewers' comments

Reviewer #1 (Remarks to the Author):

The authors combine flux tower CO₂ fluxes (GPP, RECO, NEE) in 39 sites, cambial activity in 85 sites and non-structural (NSC) carbohydrates in 57 sites representative of conifers in boreal, temperate, and Mediterranean forest stands to study the seasonal timing of plant processes, such as carbon fluxes, NSC dynamics, and wood formation. The study is motivated by the need to reconcile carbon sources and sinks controls on wood formation and therefore carbon sequestration with obvious implications for plant growth representation in terrestrial ecosystem models and global carbon cycle.

The manuscript is well written and presented, I read with a lot of interest. However, while I agree with most of what is discussed in the manuscript, I think the new results add very little and do not largely fulfill the risen expectations for a number of reasons detailed below.

Answer: We thank the reviewer for the feedback on our work. Below, we have addressed each of the points raised:

Question: The analysis of the diverse datasets that have been compiled is rather simplistic, the authors just search for the seasonal maximum of a given process/variable by fitting some non-linear function(LL 406-409) to normalize the time series. This reduces the new results to the timing of the peak of the various processes (Fig. 2, 3, 4), but with no information about their magnitude or the intra-seasonal variability of the process itself. While of course the knowledge of the DoY of maximum activity in so many case studies is relevant, it is only a very partial information. For instance, I am wondering, if it would not have been better to fit a CDF at least for some of the variables, and discuss the timing of the 10, 25, 50, 75, 90 percentiles, rather than the "maximum". It would have been more representative of the overall seasonality of the process; the maximum is really a specific information. The outcome of the simplistic analysis is that conclusions are remarking largely well-known seasonal sequences as that NEE peaks before GPP or the well-known order of cambial activity: cell division, cell enlargement and cell wall thickening and lignification (reviewed by some of the authors themselves, Rathgeber et al 2016). As a matter of fact, most of these processes have a strong temperature signal and it is not surprising to see correlation across sites that have different average temperatures, the temperature effect itself will shift all of them later in the season in colder sites (Fig. 4). Most of what is discussed inside the article is correct, but it is a review based on literature and previous knowledge (e.g., LL 250-260, LL 323-330 and many others), with very little or no-addition from the results presented here. In other words, most of the article could have been written without the results of Fig. 2, 3 and 4.

Rathgeber, C. B., Cuny, H. E., & Fonti, P. (2016). Biological basis of tree-ring formation: a crash course. *Frontiers in Plant Science*, 7, 734.

Answer: We have undertaken several actions to address the reviewer's comments. However, we will provide a more comprehensive response in the following comment/answer section to address concerns related to the magnitude of processes and the significance of our work.

We agree with the reviewer's suggestion to include a description of different percentiles to enhance the depth of our results and offer a more detailed view of seasonal patterns. In response, we have calculated and incorporated the 10th, 25th, 50th, 75th, and 90th percentiles for each curve related to cambial activity and wood formation phenological stages (i.e., cell enlargement and cell wall thickening and lignification), as well as for the curves representing various carbon fluxes (i.e., NEE, GPP, and RECO) for FluxNet sites. In this context we calculated the percentiles for both the ascending and descending portions of the curves, acknowledging that our regressions account for non-zero skewness. To facilitate comparisons of seasonal patterns, we have also calculated the time differences between these carbon fluxes and wood formation phenological stages. This analysis contributed significantly to our manuscript, offering a more detailed insight into the synchronization during peak periods between cell differentiation and carbon assimilation, but also highlighted the asynchronization of the two processes during off-peak periods.

Consequently, we have thoroughly revised and updated Figure 3 to present these additional results, extending beyond the peaks, to provide a more comprehensive picture of the seasonal patterns. Thus, we have also modified the caption as follows: *“Differences (i.e., subtraction) between the timing of different percentiles and peak (i.e., 100th percentile) of C fluxes, i.e., NEE, GPP, RECO and those of phenological phases of wood formation, i.e., cambial activity, cell enlargement and cell wall thickening and lignification in boreal, temperate and Mediterranean biomes. Negative deltas indicate that C fluxes occurred earlier, while positive deltas that C fluxes occurred later, relative to wood formation, for each percentile shown in the figure.”*.

The description of this further analysis has been reported in material and methods section at lines 506-513: *“Each curve estimated the timing of the maximum value (or minimum while considering soluble sugar concentrations). In addition to this, fitted curves pertaining to the seasonal pattern of wood formation (i.e., cambial activity, cell enlargement, and cell wall thickening and lignification) and carbon fluxes derived from FluxNet data (i.e., NEE, GPP, and RECO) were utilized to estimate the timing of key percentiles, namely the 10th, 25th, 50th, 75th and 90th percentiles, for both the ascending and descending portions of the curves.”*

We presented the results related to this analysis in result and discussion sections at lines 327-341: *“Photosynthesis and cell differentiation are synchronized across a wide spatial scale (Fig. 3), probably because the two processes occur during the time window when environmental conditions are optimal for both. By considering the 10th percentiles of the fitted curves, the onset of photosynthesis occurs earlier (55 days, average across biomes) than the onset of cell differentiation during wood formation (i.e., onset of cell enlargement stage) (Fig. 3, Table S7). Conversely, considering the 10th percentiles of the descending portions of the curve the ending of photosynthesis occurs later (33 days, average across biomes) than the ending of cell differentiation (i.e., ending of cell wall thickening and lignification stage) (Fig. 3, Table S7). As*

we progress towards the peaks of these processes, the time lag between GPP and wood formation gradually diminishes (Fig. 3, Table S7). During the period of maximum activity (75th percentile, ascending portion of the curves), the time gap between GPP and the cell enlargement stage narrows to an average of 18 days across biomes, and 12 days across biomes between GPP and the cell wall thickening and lignification stage, in the descending portions of the curve (Fig. 3, Table S7).”

We have now included a supplementary table (Table S7; page 42) that provides a comprehensive list of the time differences between C fluxes and wood formation phenological phases. This table also serves as the dataset used to create Figure 3 in the main text.

Question: The presented results are all based only on temporal sequences, but there is no assessment of the process magnitude. In my view, reporting values of the different fluxes in gC m⁻² day or storages in gC m⁻² is fundamental, if we want to advance the field and really address the challenges discussed by the authors (LL 162-166, 190-197, 380-381). I understand that for some variables (e.g., NSC dynamics, cambial activity) scaling values to forest stand might be particularly complex, but even with uncertainties we should aim at that. There is a value in knowing when certain processes are taking place in the growing season, however, it is their magnitude which controls the relative importance of sources and sink controls. One can have NPP and growth fully synchronized seasonally, but as wood growth is only a fraction of NPP (e.g., Wolf et al 2011), it is much more difficult to discuss their direct link. In other words, we need to understand seasonally where the gC m⁻² are allocated, stored, and translocated. For instance, I found the article of Gough et al 2009 focusing on a single site and estimating actual stand scale magnitudes of C allocation much more useful than looking at hundreds of sites in some normalized fashion. This is what I was expecting from the title and introduction of such an article.

Finally, the intra-annual perspective, if analyzed in terms of actual magnitude, can provide a lot of information on the mechanisms of carbon allocation, I agree with LL 188-190. However, when one wants to quantify carbon sequestration on wood biomass (a major motivation, LL 161-169), it is the correlation at the annual or multi-annual scale (if we account for previous year legacy and storage) between GPP and wood formation that should matter. If some C meant for wood is partially stored during the growing season and allocated to wood a few weeks later is surely relevant for process understanding, but it will not modify the overall C budget and it will unlikely require substantial changes to current models.

Gough C.M., Flower, C.E., Vogel, C.S., Dragoni, D., Curtis, P.S., 2009. Whole-ecosystem labile carbon production in a north temperate deciduous forest. *Agric. Forest Meteorol.* 149, 1531–1540.

Wolf, A., Field, C. B., & Berry, J. A. (2011). Allometric growth and allocation in forests: a perspective from FLUXNET. *Ecological Applications*, 21(5), 1546-1556.

Answer: We have considered both comments in the next answer to provide a more comprehensive explanation.

We agree with the reviewer that rightly emphasizes the importance of studying the magnitude of the processes under investigation and quantifying the amount of carbon assimilated, allocated, stored, and translocated. However, it is crucial to acknowledge that the study of temporal

patterns in these processes has been and continues to be a topic of debate in the literature. This is largely due to the emergence of contrasting results, which can be attributed to the utilization of various approaches employed to address the complex issue of assessing temporal relationships between these processes. These discrepancies are evident in both localized studies and data syntheses, as we have indicated in line 181-186 of the manuscript: "*Nowadays, the possibility to use direct measurements and multi-year records of ecosystem C fluxes from eddy-covariance (EC) stations in forests has greatly increased the potential of assessing the association between C fluxes at ecosystem scale with wood production. However, these analyses have proposed different or even contrasting conclusions. While some studies showed strong correlations between source and sink activities¹⁵⁻²⁰, others lacked in significant results²¹⁻²⁵.*".

Hence, considering the importance of identifying temporal relationships between these processes, as acknowledged by the reviewer, the primary objective of this work was to provide a comprehensive description of phenological patterns and their relationships. In doing so, we aimed to clarify that to assess temporal relationships among processes, and potentially their magnitudes, it is imperative to employ monitoring techniques that enable quantification and observation at an intra-annual scale. Notably, techniques such as eddy covariance flux data and xylogensis assessments are well-suited for this purpose. This also represents the novelty of our work, as prior studies conducted on a broad spatial scale have primarily addressed the issue by concentrating on interannual patterns. However, investigations conducted at an annual resolution cannot effectively capture the nuances of physiological processes occurring throughout the growing season. In this context, the present work holds clear potential to establish a fundamental baseline and a comprehensive comparative framework that is currently missing. Local studies can undoubtedly benefit from this work by using it as a foundational reference point. This point has been significantly clarified in the revised version of the manuscript, particularly in the introduction section at line 187-206: "*The contrasting results reported by the literature in the last decades¹⁵⁻²⁵ could be explained by the different approaches employed to address the complex issue of assessing temporal relationships between these processes. In this context, some studies have explored temporal relationships between source and sink activities, focusing on interannual patterns. At a global scale, in particular, eddy covariance GPP records have been shown to be largely decoupled from tree growth at the inter-annual time scale²³. Several reasons were proposed to explain this asynchrony, e.g., stored carbohydrates may provide much of the C necessary for growth during certain growth stages, and the seasonal dynamics of GPP and wood formation may substantially differ from each other. Nevertheless, it is crucial to highlight the importance of precisely defining the temporal resolution when investigating the temporal dynamics of specific processes. Indeed, studies conducted at annual resolution cannot assess physiological processes occurring during seasonal growth and the underlying mechanisms may significantly vary within and across years. This issue should be addressed with observations performed at higher temporal resolution because intra-annual growth-related physiological processes may demonstrate a buffering effect, possibly desynchronizing source and sink activities. A detailed analysis of these processes and the examination of their seasonal patterns at intra-annual scale may provide a more comprehensive understanding of the C cycle in forest ecosystems and quantify the degree of synchrony between C sources and sinks at different spatial scales, from stand to ecosystem.*".

To emphasize this point further, we have included the following text in the last paragraph of the

discussion at line 428-430: “Numerous sites are currently equipped for monitoring and measuring carbon fluxes in various forest ecosystems. However, within the last decade, only a small percentage of these sites have been monitoring wood formation on an intra-annual scale.”

Regarding the assessment of magnitude, we concur on the significance of quantifying the actual magnitudes of these processes. However, in the context of this work, achieving this quantification is firstly constrained by the availability of data. To estimate carbon content, we would require species-specific allometric equations and wood density measurements, also incorporating growth measurement such as shoot elongation and radial growth, at least for the years under study. Most of these data are not actually available for most of the study sites used in this work. Moreover, assuming the availability of these data, integrating the actual magnitudes of these processes alongside the phenological patterns would have likely overwhelmed the scope of this work that is mainly focused on phenological patterns.

In light of the reviewer's comments, we however believe it is worthwhile to further explore and emphasize the point he/she raised in the main text of the manuscript. In addition to the modifications presented above, we included in our discussion at lines 370-481 the following paragraph: “*The limited availability of sites with concurrent measurements of C fluxes, NSC concentrations, and wood phenology has hindered our ability to quantitatively explore the relationships between these three categories of processes. The extent to which C sources and sinks are closely linked is fundamentally determined by the magnitude of C fluxes moving between different compartments, which are exceedingly difficult to measure. However, given the ongoing debates surrounding the significance of potential drivers, temporal sequences, and the level of correlation among these processes at larger temporal and spatial scales, we believe that our temporal analysis provides a unique broad picture of the overall coordination of source and sink activities (and ultimately of C sequestration in wood) at large scales. Consequently, our study serves as a foundation to deepen our understanding of the consequences of climate changes on C sequestration in Northern hemisphere forests, along with exploring potential biome-specific responses.*”

Moreover, we have also revised the last paragraph of the discussion (lines 430-440), which now reads as follows: “*Incorporating detailed intra-annual observations of NSC pools, which act as a buffer between atmospheric C flux measurements (e.g., eddy covariance GPP and RECO) and intra-annual assessments of forest biomass growth (i.e., wood formation monitoring and intra-ring analysis of wood anatomy), can facilitate the complex task of reconciling, and ultimately clarifying, the temporal and functional relationships between C uptake and long-term C sequestration. This, in turn, will enable the examination of the timing and magnitude of C transfer processes and C use efficiency across various spatial and temporal scales. Such an analysis is a crucial step to reduce the sources of uncertainty in global vegetation models and achieve a deeper understanding of the C cycle at global scale.*”

15. Xu, K. *et al.* Tree-ring widths are good proxies of annual variation in forest productivity in temperate forests. *Sci. Rep.* **7**, 1–8 (2017).
16. McKenzie, S. M., Pisaric, M. F. J. & Arain, M. A. Comparison of tree-ring growth and eddy covariance-based ecosystem productivities in three different-aged pine plantation forests. *Trees - Struct. Funct.* **35**, 583–595 (2021).
17. Metsaranta, J. M., Mamet, S. D., Maillet, J. & Barr, A. G. Comparison of tree-ring and eddy-covariance derived annual ecosystem production estimates for jack pine and trembling aspen forests in Saskatchewan,

- Canada. *Agric. For. Meteorol.* **307**, 108469 (2021).
18. Teets, A. *et al.* Linking annual tree growth with eddy-flux measures of net ecosystem productivity across twenty years of observation in a mixed conifer forest. *Agric. For. Meteorol.* **249**, 479–487 (2018).
 19. Tei, S. *et al.* Strong and stable relationships between tree-ring parameters and forest-level carbon fluxes in a Siberian larch forest. *Polar Sci.* **21**, 146–157 (2019).
 20. Puchi, P. F. *et al.* Revealing how intra- and inter-annual variability of carbon uptake (GPP) affects wood cell biomass in an eastern white pine forest. *Environ. Res. Lett.* **18**, 024027 (2023).
 21. Rocha, A. V., Goulden, M. L., Dunn, A. L. & Wofsy, S. C. On linking interannual tree ring variability with observations of whole-forest CO₂ flux. *Glob. Chang. Biol.* **12**, 1378–1389 (2006).
 22. Delpierre, N., Berveiller, D., Granda, E. & Dufrêne, E. Wood phenology, not carbon input, controls the interannual variability of wood growth in a temperate oak forest. *New Phytol.* **210**, 459–470 (2016).
 23. Cabon, A. *et al.* Cross-biome synthesis of source versus sink limits to tree growth. *Science (80-.)*. **376**, 758–761 (2022).
 24. Oddi, L. *et al.* Contrasting responses of forest growth and carbon sequestration to heat and drought in the Alps. *Environ. Res. Lett.* **17**, 108030 (2022).
 25. Krejza, J. *et al.* Disentangling carbon uptake and allocation in the stems of a spruce forest. *Environ. Exp. Bot.* **196**, 104787 (2022).

Specific Comments.

Question: P. 10 LL 332-333. I think this sentence is incorrect and a much more nuanced approach to the fact that CO₂ fertilization might induce considerable changes, in various processes, which have been widely documented (e.g., Walker et al 2021), should be made.

P. 11 LL 365. Here, the text contradicts with LL 332-333.

Walker, A. P., De Kauwe, M. G., Bastos, A., Belmecheri, S., Georgiou, K., Keeling, R. F., ... & Zuidema, P. A. (2021). Integrating the evidence for a terrestrial carbon sink caused by increasing atmospheric CO₂. *New phytologist*, 229(5), 2413-2445.

Answer: According to the reviewer's comment the above-mentioned sentence has been removed.

Question: P. 11. LL 371-374 This is largely repetitive of what has been written before.

Answer: The reviewer's observation is right. Indeed, we provided a concise summary of the key points presented in the preceding paragraph, intending to assist readers in comprehending the overall flow of ideas presented in the manuscript's conclusion. To clarify this, we explicitly indicated in the text that the sentence served as a summary by commencing it with "In summary, [...]"

Reviewer #2 (Remarks to the Author):

This study is very novel and will certainly have a high impact in the field, since it fills a gap of understanding carbon (C) cycle with a multi-disciplinary approach considering all different factors around the seasonal C assimilation by plants.

The work was performed with conifers, but can in the future be expanded to angiosperms and help us understand forest C assimilation dynamics also in the tropics, which contain the largest biodiversity in the planet.

Question: In this specific study, however, why were only conifers included? They represent a low diversity in terms of total trees in the planet, although of considerable biomass (Siberian Taiga.)

Also, for the future I think it would be invaluable to include conifers growing in tropical regions (in mountains, for instance). I think that would have given another twist to the work, since the climatic seasonality would have been completely different.

Some sentences, like the very first one from Intro, would benefit from citations.

I don't have any major criticism for the article, and I think it is a good fit for the journal.

Answer: We appreciate the reviewer's positive feedback on our work. The current focus on conifers in this study is primarily due to the availability of data related to wood formation. Historically, research on xylogenesis has predominantly involved coniferous species. However, in recent years, there has been a growing number of studies that include angiosperms. Regarding the exclusion of sub-tropical and tropical sites, while we did have access to some data, they were clearly not enough for the representation of the respective biomes.

It's important to note that this project is an ongoing endeavor, we will likely be able to expand and diversify the dataset in the next years to encompass a broader range of species and biomes, including angiosperms and sub-tropical/tropical sites.

We finally added references in introduction at lines: 147, 160, 172, 195.

Reviewer #3 (Remarks to the Author):

This study uses a dataset of seasonal components of vegetation productivity and wood formation to quantify variation in timing of these processes and obtain insights into the dependencies of these processes. Specifically, the study provides the insight that wood formation depends on sugars produced by photosynthesis in an indirect way. That is: the analyses of timing of these processes, plus those in starch, suggest that sugars produced are first stored before being used in wood formation. This analysis provides a valuable addition to the literature on source-sink limitation on woody growth, mostly because it illustrates how source and sink limitation vary over the season. The study is conducted at a remarkably large scale, in three NH climate zones.

Answer: We thank the reviewer for the positive feedback on our work. In response to the reviewer's comments, we have provided a comprehensive and detailed response in the sections above.

I have four main comments:

Question: The manuscript lacks clear questions or hypotheses on the expected timing of the studied processes. Instead, only objectives/aims are mentioned. At times, I got the impression that the study is quite descriptive. I would advice the authors to formulate hypotheses, e.g., on

the order of processes, and differences between biomes. This will also help selecting results to be shown.

Answer: According to the editor comments we did not arise any new hypothesis at this stage for the present work. However, to address reviewer comment we enhanced the clarity of our objectives and effectively stated our main original hypothesis at lines 219-222: “Given the high C-demanding nature of wood formation, we hypothesize that a synchronization should exist between the seasonal peaks in carbon assimilation and cell differentiation during wood formation.”

Moreover, to better underscore the novelty of our work and enhance the clarity of the introduction along with the presented problem, we have completely reformulated and expanded the paragraph at lines 193-202 as it follows: “The contrasting results reported by the literature in the last decades¹⁵⁻²⁵ could be explained by the different approaches employed to address the complex issue of assessing temporal relationships between these processes. In this context, some studies have explored temporal relationships between source and sink activities, focusing on interannual patterns. At a global scale, in particular, eddy covariance GPP records have been shown to be largely decoupled from tree growth at the inter-annual time scale²³. Several reasons were proposed to explain this asynchrony, e.g., stored carbohydrates may provide much of the C necessary for growth during certain growth stages, and the seasonal dynamics of GPP and wood formation may substantially differ from each other. Nevertheless, it is crucial to highlight the importance of precisely defining the temporal resolution when investigating the temporal dynamics of specific processes. Indeed, studies conducted at annual resolution cannot assess physiological processes occurring during seasonal growth and the underlying mechanisms may significantly vary within and across years. This issue should be addressed with observations performed at higher temporal resolution because intra-annual growth-related physiological processes may demonstrate a buffering effect, possibly desynchronizing source and sink activities. A detailed analysis of these processes and the examination of their seasonal patterns at intra-annual scale may provide a more comprehensive understanding of the C cycle in forest ecosystems and quantify the degree of synchrony between C sources and sinks at different spatial scales, from stand to ecosystem.”

15. Xu, K. *et al.* Tree-ring widths are good proxies of annual variation in forest productivity in temperate forests. *Sci. Rep.* **7**, 1–8 (2017).
16. McKenzie, S. M., Pisaric, M. F. J. & Arain, M. A. Comparison of tree-ring growth and eddy covariance-based ecosystem productivities in three different-aged pine plantation forests. *Trees - Struct. Funct.* **35**, 583–595 (2021).
17. Metsaranta, J. M., Mamet, S. D., Maillet, J. & Barr, A. G. Comparison of tree-ring and eddy-covariance derived annual ecosystem production estimates for jack pine and trembling aspen forests in Saskatchewan, Canada. *Agric. For. Meteorol.* **307**, 108469 (2021).
18. Teets, A. *et al.* Linking annual tree growth with eddy-flux measures of net ecosystem productivity across twenty years of observation in a mixed conifer forest. *Agric. For. Meteorol.* **249**, 479–487 (2018).
19. Tei, S. *et al.* Strong and stable relationships between tree-ring parameters and forest-level carbon fluxes in a Siberian larch forest. *Polar Sci.* **21**, 146–157 (2019).
20. Puchi, P. F. *et al.* Revealing how intra- and inter-annual variability of carbon uptake (GPP) affects wood cell biomass in an eastern white pine forest. *Environ. Res. Lett.* **18**, 024027 (2023).
21. Rocha, A. V., Goulden, M. L., Dunn, A. L. & Wofsy, S. C. On linking interannual tree ring variability with observations of whole-forest CO₂ flux. *Glob. Chang. Biol.* **12**, 1378–1389 (2006).
22. Delpierre, N., Berveiller, D., Granda, E. & Dufrêne, E. Wood phenology, not carbon input, controls the

- interannual variability of wood growth in a temperate oak forest. *New Phytol.* **210**, 459–470 (2016).
23. Cabon, A. *et al.* Cross-biome synthesis of source versus sink limits to tree growth. *Science* (80-.). **376**, 758–761 (2022).
 24. Oddi, L. *et al.* Contrasting responses of forest growth and carbon sequestration to heat and drought in the Alps. *Environ. Res. Lett.* **17**, 108030 (2022).
 25. Krejza, J. *et al.* Disentangling carbon uptake and allocation in the stems of a spruce forest. *Environ. Exp. Bot.* **196**, 104787 (2022).

Question: Sites and representativeness. A total of 180 sites is used, and for some variables (e.g., GPP) gridded products are used. Figure 1 shows there are hardly (no?) sites at which multiple processes (starch, wood formation, etc) are measured. This may have important implications for within-biome comparisons of different processes and cross-biome comparisons of the same process. For instance, if wood xylogenesis in biome X is measured at warmer sites, while starch levels at cooler sites, the difference in timing of these processes will not only be determined by the order of processes for an average tree in these sites, but also by across-site differences in climate. As far as I understand, this issue has not been accounted for. It is important to conduct a robustness test to verify whether climatic conditions within biomes differ across sites that deliver different processes. Similarly, differences in the years covered by the field studies included may also have an impact, which needs to be quantified. And, as far as I can tell, the analyses do not take into account site information as a random factor, and now mix data within and across sites. Finally, I think it would be important to show how representative sites are for the biomes studied. Is the full climate range represented?

Answer: We agree with reviewer comment that a further effort was in order to show representativeness of the sites under consideration. The first point that is worthy to be discussed is that to our knowledge no sites are currently available where eddy-covariance C fluxes, NSC, and wood formation dynamics have been measured concurrently. To be more precise, even just few sites have conjunct C fluxes and wood formation assessments. In our conclusions, we also added a paragraph at lines 387-395 to remark that this is a central research priority for future research: “*The limited availability of sites with concurrent measurements of C fluxes, NSC concentrations, and wood phenology has hindered our ability to quantitatively explore the relationships between these three categories of processes. The extent to which C sources and sinks are closely linked is fundamentally determined by the magnitude of C fluxes moving between different compartments, which are exceedingly difficult to measure. However, given the ongoing debates surrounding the significance of potential drivers, temporal sequences, and the level of correlation among these processes at larger temporal and spatial scales, we believe that our temporal analysis provides a unique broad picture of the overall coordination of source and sink activities (and ultimately of C sequestration in wood) at large scales. Consequently, our study serves as a foundation to deepen our understanding of the consequences of climate changes on C sequestration in Northern hemisphere forests, along with exploring potential biome-specific responses.*”.

To address the reviewer's comment, our initial step involved opting for a graphical representation of the sites utilized in this study based on their mean annual temperature and annual precipitation. In the revised Figure 1 of the main text, alongside the map illustrating the spatial distribution of the study sites, we have incorporated a Whittaker biome plot. This plot represents all our sites classified according to the type of data originating from each site. Additionally, in the annexes, we introduced a new figure (Figure S2, page 12) where the same Whittaker biome

plot is utilized to illustrate a comparison. This comparison contrasts our biome classification, derived from the information retrieved directly from the original study source of the data, with the Whittaker classification. While somewhat simplistic, given that the classification is based solely on two broad climatic variables, it provides an immediate representation of the climatic distribution of our study sites and show an overlap between our classification and Whittaker's one.

Secondly, to be more detailed about the classification we considered all the sites in which wood formation have been monitored and we performed an in-depth bioclimatic analysis to validate our classification according to the climatic biome and validate the main result presented in figure 4.

Precisely, as reported in the main text of the revised manuscript at lines 474-489: *“To assess the climatic differences among sites where wood formation has been monitored, we collected bioclimatic data from the CHELSA bioclimatic database V2.1^{55,56} with a spatial resolution of 30 arcseconds. Out of the 19 available bioclimatic parameters^{55,56}, we selected seven variables (Table S1), excluding those that provided overly general descriptions of climate (e.g., annual temperature and precipitation) and removing variables that were highly correlated with each other ($r > |0.7|$). Subsequently, to group the study sites based on their climate-related characteristics, we applied the Partitioning Around Medoids (PAM) clustering algorithm, which is an extension of the k-means clustering algorithm⁵⁷. To determine the optimal number of clusters, we utilized the Within-Sum-of-Squares method (WSS), which minimizes the distance between points within each cluster. Therefore, we determined that four clusters were the optimal choice for grouping the wood formation study sites. We then conducted a Principal Component Analysis (PCA) on the bioclimatic variables to determine the climatic classification of our 81 sites. Pearson's correlation coefficient was employed to identify the climatic factors that influenced the ordering of wood formation study sites by principal components (Table S1).”*

Consequently, we presented the results of the bioclimatic analysis in the annexes at pages 24 and 25: *“The PCA of the 81 sites in which wood formation assessments were carried out yielded eight components, of which two were significant, collectively accounting for 79.63% of the variance (Fig. S3 and Table S1). The first component (PC1), which explained 49.90% of the variance (Table S1), showed a strong correlation ($r > |0.7|$) with the mean temperatures of the driest, warmest, and coldest quarters (bio9, bio10, and bio11, respectively), as well as the precipitation of the warmest quarter (bio18) (Table S1). The second component (PC2), responsible for explaining 29.73% of the variance (Table S1), displayed a strong correlation ($r > |0.7|$) with temperature seasonality (bio4). It also showed moderate correlations ($|0.5| > r > |0.7|$) with the mean temperature of the coldest quarter (bio11), precipitation seasonality (bio16), and precipitation of the driest and warmest quarters (bio17 and bio18, respectively) (Table S1). Partitioning Around Medoids clustering algorithm facilitated the distinction of four bioclimatic clusters within the Northern Hemisphere, with a clear border between climatic biomes (Fig. S3). The Mediterranean cluster showed direct correlations with the mean temperature of the driest, warmest, and coldest quarters (bio9, bio10, and bio11, respectively) while inversely correlating with precipitation seasonality (bio16) and precipitation of the driest and warmest quarters (bio17 and bio18, respectively) (Fig. S3). The sites located in temperate biomes belonged to two main clusters, with the colder sites being directly correlated with*

precipitation seasonality (bio16) and precipitation of the driest and warmest quarters (bio17 and bio18, respectively), and inversely correlated with the mean temperature of the driest, warmest, and coldest quarters (bio9, bio10, bio11, respectively) (Fig. S3). The central temperate cluster was associated with most bioclimatic variables used in the analysis. Lastly, the boreal cluster was primarily determined by temperature seasonality (bio4) (Fig. S3)."

The decision to present the results of this additional analysis in the annexes is primarily driven by the consistency of the main findings of the present work considering both the classifications. As output of the bioclimatic analysis, we examined four distinct climatic groups. We identified a difference within the temperate biome, which separated into two distinct subgroups. This outcome was somewhat predictable, given that the temperate biome, among those considered in this study, encompasses a broader variability in climatic conditions compared to boreal and Mediterranean biomes, as also evident in the Whittaker biome plot. According to our analysis, these subgroups primarily represent warmer and colder sites within the temperate biome.

To present the results with the most fitting classification, we conducted all subsequent analyses bringing to our main result, as detailed in the previous version of the manuscript. Overall, as already mentioned, we found consistent synchronization between the peak in Gross Primary Productivity (GPP) and the phenological stages of wood differentiation, as well as consistent time gaps between different groups and biomes. Consequently, to present our primary findings more clearly, we chose to maintain the classification according to biome in the main text. However, we shared all the results, figures and tables related to cluster analysis, Principal Component Analysis (PCA), and regressions in annexes, specifically on pages 8, 9, 24, 25, and 52.

⁵⁵ Karger D.N. *et al.* Data from: Climatologies at high resolution for the earth's land surface areas. *EnviDat* (2018).

⁵⁶ Karger, D.N. *et al.* Climatologies at high resolution for the Earth land surface areas. *Scientific Data*. 4 170122 (2017)

⁵⁷ L. Kaufman & P.J. Rousseeuw. *Clustering by means of Medoids*. Y. Dodge (Ed.), Statistical Data Analysis Based on the L1-Norm and Related Methods, North-Holland, pp. 405-416 (1987).

Question: In the main text, very little information is provided about the uncertainty of the analyses, or variation within and across biomes. No mention is made of the explained variance (or RSE) of the statistical models. Variation across sites is only shown in Figure 4, but not in 1-3. And in the text, information on peak days, or duration of processes is given without any information on variation, ie. no standard deviations or errors or CI95 are provided.

Answer: According to the reviewer comment we added to the main text information about the uncertainty of fitting curves at lines 235-240: "The seasonal dynamics of wood formation (i.e., cambial activity, cell enlargement, and cell wall thickening and lignification), NSC (i.e., starch and soluble sugars in needles, stem and roots), and carbon fluxes (i.e., NEE, GPP and RECO) were fitted with skewed normal distribution or V-type exponential curves. All curves were significant ($p < 0.05$) with a residual standard error (RSE) ranging between 0.09 and 0.27 (Supplementary Material 3 - Table S3)". Contextually, in annexes at page 27 is also presented a table resuming the RSE for all individual curves as well as the parameters estimate (peaks

included) and respective standard errors.

Question: I think the presentation of the results in Figures can be substantially improved. In Fig 1: the map could show forest cover in the background (or duration of growing season?), could be limited to NH sites, and it would have been good to plot sites in climate space as well, so show representativeness. IN Figure 2, the same information is provided on the left and right panels. Also a grouping of processes (related to photosynthesis, to storage or to wood formation) by colouring lines would be helpful. And, finally on this Figure, I thought that actual fitted curves in the SI were more informative than the way results are presented in Fig 2, because one can see the intensity of the processes over time. In Fig 3, I do not get what is shown, and the caption was not really helpful in explaining.

Answer: We made an effort to address all the suggestions provided by the reviewer.

In the case of Figure 1, we opted not to introduce additional visual elements to the map to preserve its readability and ensure the visibility of all site locations. However, we have integrated a Whittaker biome plot in Figure 1, illustrating the mean annual temperature (°C) and mean annual precipitation (cm) for all 177 study sites.

Regarding Figure 2, while it is true that the left and right panels convey the same information, they serve different purposes. The left panel is designed to facilitate comparisons within biomes, while the right panel aids in comparing across biomes. This approach is also reflected in the results section of the main text. To enhance clarity, we have color-coded the processes by biome, using gradients to represent flux, NSC, and wood formation processes. While we acknowledge that actual curves can provide more detail, incorporating 36 fitted curves made summarizing the results a complex task when aiming to use the fitted curves directly. However, to clarify this point also for readers, we modified the figure 2 caption as it follows: “Fig. 2 *Peak (dot or triangle) and period of maximum activity (horizontal lines) for C fluxes, NSC dynamics and wood formation phenological phases. Minimum concentration is shown for soluble sugars. On the left, for each biome, the sequence of temporal consecutive events during the growing season is shown. On the right, the difference among biomes for each event is shown. NEE, GPP and RECO represent the Net Ecosystem Exchange, Gross Primary Production and Ecosystem Respiration. ST and SS represent starch and soluble sugars, respectively. Cell WTL represents the phenological stage of cell wall thickening and lignification.*”

Finally, in response to the reviewer's suggestion, we completely redesigned the third figure. This figure now illustrates the differences in timing between the 10th, 25th, 50th, 75th, 90th percentiles, and the peak among C fluxes and wood formation phenological phases in the three biomes under consideration.

We have now included a supplementary table (Table S7) that provides a comprehensive list of the time differences between C fluxes and wood formation phenological phases. This table also serves as the dataset used to create Figure 3 in the main text.

Two minor comments:

Question: In the text, the term mismatch is used, which -- to me -- suggests that something is

going 'wrong', but that's clearly not the case. Processes are out of sync, but they are still connected functionally.

Answer: We agree with the reviewer that mismatch in this context can be misleading for readers. Accordingly, we substitute the noun “mismatch” with “asynchrony” and, when used as a verb, with “desynchronize”.

Question: The focus in the text and Figures is on comparing the peaks in the processes, but why not the onset of processes, or the time at which 90% of the seasonal total is reached? Is the analyses of peaks really the best way to get insights on seasonal dynamics of source-vs-sink limitation?

Answer: To address the reviewer’s comment, we have calculated and incorporated the 10th, 25th, 50th, 75th, and 90th percentiles for each curve related to cambial activity and wood formation phenological stages (i.e., cell enlargement and cell wall thickening and lignification), as well as for the curves representing various carbon fluxes (i.e., NEE, GPP, and RECO) for FluxNet sites. To facilitate comparisons of seasonal patterns, we have also calculated the time differences between these carbon fluxes and wood formation phenological stages. This analysis contributed significantly to our manuscript, offering a more detailed insight into the synchronization during peak periods between cell differentiation and carbon assimilation, but also highlighted the asynchronization of the two processes during off-peak periods. Consequently, we have thoroughly revised and updated Figure 3 to present these additional results, extending beyond the peaks, to provide a more comprehensive picture of the seasonal patterns. Thus, we have also modified the caption as follows: *“Differences (i.e., subtraction) between the timing of different percentiles and peak (i.e., 100th percentile) of C fluxes, i.e., NEE, GPP, RECO and those of phenological phases of wood formation, i.e., cambial activity, cell enlargement and cell wall thickening and lignification in boreal, temperate and Mediterranean biomes. Negative deltas indicate that C fluxes occurred earlier, while positive deltas that C fluxes occurred later, relative to wood formation, for each percentile shown in the figure.”*

The description of this further analysis has been reported in material and methods section at lines 506-512: *“Each curve estimated the timing of the maximum value (or minimum while considering soluble sugar concentrations). In addition to this, fitted curves pertaining to the seasonal pattern of wood formation (i.e., cambial activity, cell enlargement, and cell wall thickening and lignification) and carbon fluxes derived from FluxNet data (i.e., NEE, GPP, and RECO) were utilized to estimate the timing of key percentiles, namely the 10th, 25th, 50th, 75th and 90th percentiles, for both the ascending and descending portions of the curves.”*

We presented the results related to this analysis in result and discussion section at lines 327-341: *“Photosynthesis and cell differentiation are synchronized across a wide spatial scale (Fig. 3), probably because the two processes occur during the time window when environmental conditions are optimal for both. By considering the 10th percentiles of the fitted curves, the onset of photosynthesis occurs earlier (55 days, average across biomes) than the onset of cell differentiation during wood formation (i.e., onset of cell enlargement stage) (Fig. 3, Table S7). Conversely, considering the 10th percentiles of the descending portions of the curve the ending of photosynthesis occurs later (33 days, average across biomes) than the ending of cell*

differentiation (i.e., ending of cell wall thickening and lignification stage) (Fig. 3, Table S7). As we progress towards the peaks of these processes, the time lag between GPP and wood formation gradually diminishes (Fig. 3, Table S7). During the period of maximum activity (75th percentile, ascending portion of the curves), the time gap between GPP and the cell enlargement stage narrows to an average of 18 days across biomes, and 12 days across biomes between GPP and the cell wall thickening and lignification stage, in the descending portions of the curve (Fig. 3, Table S7)."

We have now included a supplementary table (Table S7; page 42) that provides a comprehensive list of the time differences between C fluxes and wood formation phenological phases. This table also serves as the dataset used to create Figure 3 in the main text.

REVIEWER COMMENTS

Reviewer #1 (Remarks to the Author):

While I agreed with most of what is discussed in the previous manuscript, I was quite critical in a number of points related to the simplicity of the analysis, the degree of novelty beyond what we already knew, and the lack of any assessment of the magnitude of the temporal dynamics of carbon fluxes and other quantities analyzed in the article.

The authors have reworked the article to address some of my comments, for instance looking at percentiles to analyze the temporal dynamics of the different quantities I think this is valuable addition and represents a more substantial novelty in comparison to the first version. The overall novelty of the work has also been better introduced. The lack of a quantification of the magnitudes of the processes, I think it is still a limitation of this and other similar studies, but as argued by the authors this is an "exceedingly difficult" problem due to lack of appropriate data in most of the sites. Something more could have been done, but I think the additions to the discussion around this problem (LL 370-388, LL 436-440) it is probably sufficient at this stage, keeping in mind that the magnitude of the processes is what we should aim to quantify next.

Specific Comments.

P. 16 LL 314. I think it should be "later" and not "earlier".

Reviewer #3 (Remarks to the Author):

In response to my previous comments (Rev #3), the authors have conducted additional bioclimatic analyses to show consistency with grouping in Whittaker biomes. Also, Figure 3 has been improved, and the discussion on synchronization of timing and peaks is more extensive and interesting now. I am satisfied with these changes.

Yet, my comments on the need for a robustness analysis and reporting of uncertainty (second and third main comment) have not been satisfactorily addressed (nor convincingly rebutted). Without a robustness test and without showing uncertainty, it is hard to assess for a reviewer or reader to what extent the differences in timing of processes (Δd) result from the sequence of biological processes, or from differences in species / climate / vegetation across the datasets used. In other words, readers cannot determine how certain or generic the reported Δd values are (and/or how sensitive they are to the specific composition of the dataset).

In the present file we provide detailed responses to reviewers' comments and feedback. We thank the reviewers for their meticulous revision efforts, which have significantly contributed to improving the clarity and coherence of the manuscript.

Below, you will find detailed responses to each comment and feedback received:

Reviewers' comments

Reviewer #1 (Remarks to the Author):

While I agreed with most of what is discussed in the previous manuscript, I was quite critical in a number of points related to the simplicity of the analysis, the degree of novelty beyond what we already knew, and the lack of any assessment of the magnitude of the temporal dynamics of carbon fluxes and other quantities analyzed in the article.

The authors have reworked the article to address some of my comments, for instance looking at percentiles to analyze the temporal dynamics of the different quantities I think this is valuable addition and represents a more substantial novelty in comparison to the first version. The overall novelty of the work has also been better introduced. The lack of a quantification of the magnitudes of the processes, I think it is still a limitation of this and other similar studies, but as argued by the authors this is an "exceedingly difficult" problem due to lack of appropriate data in most of the sites. Something more could have been done, but I think the additions to the discussion around this problem (LL 370-388, LL 436-440) it is probably sufficient at this stage, keeping in mind that the magnitude of the processes is what we should aim to quantify next.

Answer: We thank the reviewer for the feedback on our work.

Specific Comments.

Question: P. 16 LL 314. I think it should be "later" and not "earlier".

Answer: The reviewer's observation is right, and we corrected accordingly.

Reviewer #3 (Remarks to the Author):

Question: In response to my previous comments (Rev #3), the authors have conducted additional bioclimatic analyses to show consistency with grouping in Whittaker biomes. Also, Figure 3 has been improved, and the discussion on synchronization of timing and peaks is more extensive and interesting now. I am satisfied with these changes.

Yet, my comments on the need for a robustness analysis and reporting of uncertainty (second and third main comment) have not been satisfactorily addressed (nor convincingly rebutted). Without a robustness test and without showing uncertainty, it is hard to assess for a reviewer or reader to what extent the differences in timing of processes (Δd) result from the sequence of biological processes, or from differences in species / climate / vegetation across the datasets used. In other

words, readers cannot determine how certain or generic the reported delta day values are (and/or how sensitive they are to the specific composition of the dataset).

Answer: As proposed by the reviewer, we conducted a sensitivity analysis employing random forest regression models to assess the impact of various variables (including biome, site, species, and study year) on the timing of peaks for FluxNET, FluxSat, as well as cambial and xylem phenology. A detailed account of the methodology adopted for this analysis is provided in the Materials and Methods section of the main text, specifically outlined at lines 576-594 as it follows:

“ To analyze the variability and influence of specific predictors on the timings of peaks of C fluxes and wood formation, we employed skewed normal distribution curves. These regressions delineated the seasonal patterns of cambial activity and xylem cell differentiation for each study year, species, site, and biome. The same methodology was applied to FluxNET (i.e., NEE, GPP, RECO) and FluxSat (i.e., GPP) data. For each regression, the timings of the maximum value (i.e., peak timing) of each process was extracted. A random forest regression model was utilized to quantify the relative importance of predictors in determining the peak timing for each process. For each process, we split the timings of the maximum values into a training set (80%) and a test set (20%) to assess the model performance. Five-fold cross-validation with five repetitions was employed as a resampling method to ensure more robust performance metrics. The goodness of fit for the regression models was evaluated using the coefficient of determination (R^2) for both the training and test sets, while the root mean squared error (RMSE) was employed to measure the accuracy of the models.”

After having considered the outcomes of our additional analysis, we believe that they not only served to validate our prior findings but also contributed new valuable and interesting insights. For this reason, we have decided to integrate these new results into the main text, specifically within the Results and Discussion section. Our results show that the biome and the site emerge as the primary predictors in terms of relative importance, compared to the influence of species (included only in models related to phenological timings of wood formation as C fluxes data are at ecosystem scale) and the study year. This observation contributes to further support our main conclusion regarding the observed synchronicity of peaks in ecosystem C fluxes and cell differentiation phenological phases. Indeed, this result implies that the timing of peak occurrences exhibits a higher degree of conservatism over time compared to the more dynamic patterns observed in the onset and conclusion of the same processes.

This new insight substantiates our key result indicating a partial asynchrony in source and sink timings, primarily driven by the timing of onset and conclusion rather than the timing of maximum activity. To emphasize this finding, a dedicated paragraph titled "*Variance and predictors of phenological events*" has been incorporated into the discussion section at lines 370-420 as it follows:

“To determine whether the distribution of available data for the main processes of ecosystem fluxes and wood formation affected our conclusions, we used random forest regression models to assess the relative importance of study year, site, species and biome as predictors for the peak timing of NEE, GPP and RECO. These variables explained between 24 and 44% of the variance.

Overall, the R^2 ranged between 0.72 and 0.96 for the training set and 0.49 to 0.68 for the test set (Table S8). Biome resulted as the most important predictor followed by site and study year across all models (Fig. S14). We observed the same pattern in the random forest model applied for FluxSat data (Fig. 5), where the model explained 54.06% of the variance, showing an R^2 of 0.89 for the training set and 0.66 for the test set (Table S8).

In the random forest regression models for the peak timing of cambial activity, cell enlargement, and cell wall thickening and lignification phases, the species of the monitored trees was also considered as a predictor alongside study year, site, and biome. These models explained from the 38.09 to the 43.09% of the variance, with R^2 ranging from 0.76 to 0.83 for the training set and from 0.70 to 0.83 for the test set (Table S8). The importance of the biome was confirmed, followed by site, species, and study year in each model (Fig. 5). However, in the model for cambial activity, the species showed a greater importance than the site (Fig. 5).

Biome emerged as the most influential predictor for the peak timing of both wood formation processes and ecosystem C fluxes. This result underscores the substantial role of the broader climatic context in shaping the temporal dynamics in source and sink activities. The observation that site exceeds the importance of the study year suggests a more important influence of site-specific environmental conditions in determining the temporal occurrence of seasonal peaks. This possibly implies a more conservative pattern of peak occurrences, calibrated to the local characteristics, rather than a response to annual weather variations.

A prior study focusing on conifers in cold environments showed that the rate of xylem cell production culminates around the summer solstice⁵⁰. After that date, cell production gradually decreases until ceasing. This pattern suggests that trees would have evolved by synchronizing their growth rates with day length⁵⁰. Conversely, growth reactivation (i.e., reactivation of secondary meristem) and onset of xylem cell differentiation, despite the variability within populations^{51,52}, are driven by weather conditions, mainly temperature and water availability^{4,6,11,12,43,44}.

These insights not only clarify the outcomes of our random forest regression model but substantiate the observed synchronism of peaks in ecosystem C fluxes and cell differentiation, contrasting their asynchrony during the onset and ending of the growing season. Indeed, photosynthesis experiences fewer constraints from environmental factors compared to meristematic activity and cell differentiation^{11,41,42}, resulting in the desynchronization of both onset and ending of source and sinks activities. However, considering that both the rates of photosynthesis⁵³ and xylem cell production⁵⁰ respond to day length, it is plausible that this factor predominantly governs the synchronization of source and sink peaks. Day length likely acts as a constant environmental factor over time, ensuring the convergence of a high demand with a proportionately high supply.

Finally, we emphasize the comparable significance of predictors between the site and the species. However, these conclusions are drawn from the analysis of phenological timings in conifers. Therefore, we cannot directly recognize the potential variation introduced by broadleaf species.”

In line with the incorporation of the new paragraph, we added a new Figure 5 in the main text. This figure provides a graphical representation of the relative importance of predictors from the random forest models for FluxSat, cambial activity, and xylem cell differentiation phenological

data. To complement this, a summary table, labeled Table S9, has been added to the annexes. This table resume the information and validation metrics for each individual random forest model. Furthermore, in the supplementary materials, Figure S14 has been included to illustrate the relative importance of predictors in the random forest models based on FluxNET data.

We were unable to extend the same analysis to Non-Structural Carbohydrate (NSC) concentration in the various plant organs. This limitation arises from the fragmented nature of NSC concentration monitoring, typically spanning only a few months throughout the year. Consequently, conducting regression analyses for each site, species, and study year became impossible due to data constraints. However, even in the previous versions of the manuscript, the description of phenological timing related to NSC concentration was limited to the initial set of analyses. These analyses were designed exclusively to provide a broad and general descriptive overview of the phenological sequence across the ensemble of considered processes.

REVIEWERS' COMMENTS

Reviewer #3 (Remarks to the Author):

I'm satisfied with the additional analyses conducted by the authors to address my comments (REv#3) on sensitivity. I believe these additional analyses provide necessary and helpful insights.